# Variants in *LRRC7* lead to intellectual disability, autism, aggression and abnormal eating behaviors

Members of the leucine rich repeat (LRR) and PDZ domain (LAP) protein family are essential for animal development and histogenesis. Densin-180, encoded by *LRRC7*, is the only LAP protein selectively expressed in neurons. Densin-180 is a postsynaptic scaffold at glutamatergic synapses, linking cytoskeletal elements with signalling proteins such as the α-subunit of $Ca^{2+}$/calmodulin-dependent protein kinase II. We have previously observed an association between high impact variants in *LRRC7* and Intellectual Disability; also three individual cases with variants in *LRRC7* had been described. We identify here 33 individuals (one of them previously described) with a dominant neurodevelopmental disorder due to heterozygous missense or loss-of-function variants in *LRRC7*. The clinical spectrum involves intellectual disability, autism, ADHD, aggression and, in several cases, hyperphagia-associated obesity. A PDZ domain variant interferes with synaptic targeting of Densin-180 in primary cultured neurons. Using in vitro systems (two hybrid, BioID, coimmunoprecipitation of tagged proteins from 293T cells) we identified new candidate interaction partners for the LRR domain, including protein phosphatase 1 (PP1), and observed that variants in the LRR reduced binding to these proteins. We conclude that *LRRC7* encodes a major determinant of intellectual development and behaviour.

Leucine rich repeat (LRR) and PDZ domain proteins (LAP proteins) are essential for the establishment of apical-basolateral polarity in epithelia[1]. The single LAP ortholog in *C. elegans*, *let-413*, is essential for the formation of adherens junctions and *let-413* deficient animals are not viable[2]. In mammals, the LAP protein family consists of four members. Out of these, Scribble, Erbin and Lano also support formation of epithelial polarity and have been associated with cancer development[1,3,4]. The fourth family member, Densin-180, stands out from the others due to its neuron specific expression and due to its selective enrichment in the postsynaptic density of excitatory synapses[5,6].

At synapses, Densin-180 is believed to function as a scaffold. The C-terminal PDZ domain binds to β- and δ-catenin, as well as actinin[7–10]. The linker region between LRR and PDZ domains contains binding sites for the α-subunit of $Ca^{2+}$/calmodulin dependent protein kinase II (αCaMKII) and Shank proteins, another group of postsynaptic scaffolds[10–12]. The relevance of the N-terminal LRR region is unclear. Densin-180 deficient mice display deficits in synaptic plasticity, in particular in long-term depression (LTD), accompanied by memory deficiency. In addition, Densin-180 deficiency in mice is associated with a remarkable aggressive behaviour[13,14].

Though defects in several genes coding for interaction partners of Densin-180 (*CTNNB1*; coding for β-catenin; *CTNND2*; coding for δ-catenin; *SHANK2*, *SHANK3*) are associated with neurodevelopmental disorders (NDD) in humans[15–18], *LRRC7* has not yet been firmly established as an aetiological gene in humans. However, our recent genetic association analysis of 77,539 participants in the 100,000 Genomes Project (100KGP) found that high-impact rare variants in *LRRC7* are

---

✉e-mail: Kreienkamp@uke.de

associated with intellectual disability (ID)[19]. In addition, three individual patients with truncating variants in *LRRC7* were observed in earlier large scale exome sequencing studies of NDD patients[20–23].

Here, we identify a group of 33 patients (including one of the previously published cases[20]), each of whom carries a rare missense or predicted loss-of-function (pLoF) variant in *LRRC7*. Patients are affected by a syndrome characterised by ID, autism, attention deficit hyperactivity disorder (ADHD) and other behavioural features, including aggressiveness and impulsivity. We provide statistical and experimental evidence that the *LRRC7* variants are causal for this syndrome. Given that several of the missense variants are in the sequence encoding the LRR domain, we provide data suggesting a function of this domain in protein interactions. We demonstrate that in vitro systems such as 293T cells, the LRR domain of Densin-180 may bind to cytoskeletal proteins such as MACF1, but also to the catalytically active subunits of protein phosphatase 1 (PP1). This suggests that Densin-180 contributes to the scaffolding of kinase/phosphatase signalling modules, which are likely to be disrupted in patients with reduced *LRRC7* expression, or with certain *LRRC7* missense variants.

## Results

### Identification of *LRRC7* variants in patients with a neurodevelopmental disorder

We previously observed a high posterior probability of association (PPA) between each of 87 genes and at least one of 52 disease classes within the 100KGP Disease Group classification Neurology and neurodevelopmental disorders[19]. All but three of the 87 associations mapped to known human disorders, according to the PanelApp gene panel database[24]. The three previously unrecognised associations corresponded to *LRRC7*, *ARPC3* and *RAB3A*. Here we expand on the association between high-impact variants in *LRRC7* and the disease class Intellectual Disability, which was assigned to 5529 of the 29,741 probands in the collection[19]. Out of the 42 genes associated with this class, *LRRC7* ranked 19th, with a PPA almost equal to 1 ($1.0 – 9.3 \times 10^{-6}$) and second only to *SHANK3* within genes coding for synaptic proteins (Fig. 1A). This association was primarily due to the presence of eight index cases, each with a different pLoF variant in *LRRC7* (four nonsense, three splice site and one frame shift) (Fig. 1B, pedigrees A and C–I). There was a further pedigree in which the index case harboured a monoallelic nonsense *LRRC7* variant and had been enrolled with the Specific Disease Early onset and familial Parkinson's disease (EOPD; Fig. 1B, pedigree B).

After establishing a strong statistical association of *LRRC7* with ID, we used a parallel, multi-centric effort to confirm this association and to expand the phenotypic spectrum. In this replication collection a larger number of patients with variants in *LRRC7* was identified, using a combination of exome sequencing and panel-based sequencing of candidate genes (pedigrees J to # in Fig. 1B). None of the variants detected here is listed in the gnomAD database, with the exception of the truncating p.Val1065Serfs*5 variant (2 cases in gnomAD v4.1.0), suggesting that they are extremely rare in the general population. Information exchange between labs was enabled through the GeneMatcher website[25], which in May 2023 listed 51 cases with *LRRC7* variants. Contributors to the *LRRC7* GeneMatcher list were contacted regularly, and all cases from responding labs are listed here.

Phenotypic analysis using human phenotype ontology (HPO) terms found that in addition to ID (81%), the affected individuals with pLoF *LRRC7* variants displayed neurodevelopmental delay (84%), delayed development of speech (81%), delayed motor development (45%) and global developmental delay (30%) (Fig. 1C). Several patients were affected by autism spectrum disorder (ASD; 24%) as well as ADHD (24%). Furthermore, atypical behaviour (90%) in terms of selective mutism and impulsive and aggressive or auto-aggressive behaviour was reported, pointing to a neurobehavioral phenotype. Aggression was observed equally in male and female patients (8 patients, out of 22

individuals in the replication cohort). Four patients with truncating variants in the central part of the mRNA/protein (exons 20/21) showed hyperphagia due to reduced satiety, leading to class II obesity in three of them. However, the body mass index of others was normal. Varying types of seizures were reported in six patients; and five patients exhibited moderate brain malformations upon MRI. Detailed clinical data can be found in the Supplementary Data 1, Supplementary Data 2 and in the case descriptions in the Supplementary Information file.

In at least eight cases, the variants in *LRRC7* were inherited from their parents; five of these were classified as unaffected by the enrolling clinicians, indicating that *LRRC7* variants may display incomplete penetrance. On the other hand, there are several cases where parents are affected by ID similar to their offspring, but are not available for testing (pedigrees R, S and V). In two cases, parents carrying the *LRRC7* variant were similarly affected as their offspring (pedigrees X, Y), whereas in pedigree Z the parent carrying the LRRC7 variant displayed a different neuropsychiatric phenotype than her child. In at least eleven cases where both parents were available for analysis, we determined that the respective variant appeared de novo.

Nonsense, splice site and frameshift variants and a 99 kB deletion encompassing the last exon were observed as truncating variants. As these lead to truncated proteins, rapid mRNA degradation due to nonsense mediated mRNA decay is likely. In addition, we identified eight patients with missense variants. All probands were heterozygous for the respective *LRRC7* variant, with the exception of one case (pedigree A, Fig. 1). Here, two nonsense variants were found. As these are rather far apart in their genomic location, and as parents are unknown due to adoption, we currently cannot determine whether they are from the same or from different alleles. However, it should be noted that the p.Gln1375* variant (as well as the p.Asn1377Thrfs*13 variant from pedigree I) resides in exon 23. Expression data from rat and mouse brain indicate that this exon is skipped in a significant portion of transcripts, due to an alternative splicing process[26,27]. Thus, phenotypic consequences of variants in this exon may be less severe.

We focused on those eight cases carrying missense variants, with the goal to understand which specific aspect of the encoded Densin-180 protein is important for normal brain function. As a LAP protein, Densin-180 consists of an N-terminal LRR region, followed by a long linker region and a C-terminal PDZ domain (Fig. 2A). All patient derived missense variants change residues in the LRR domain (p.L65S; p.N94S, 2 cases; p.C106Y; p.A193P; p.L221M, p.L296P), with the exception of one variant in the PDZ domain (p.L1567Q).

Structural and sequence analysis indicated that the missense variants lead to changes in highly conserved positions in the consensus sequence of a leucine rich repeat (Fig. 2B, C). Leu65 is part of the hydrophobic core of the first leucine rich repeat of Densin-180; in fact, leucine at this position is part of the consensus sequence for LRRs, and a leucine or isoleucine is present in this position in 15 out of 17 repeats in Densin-180. It is likely that a hydrophilic serine at this position will destabilise folding of the initial LRRs of the protein. Similarly, Asn94 is at a conserved position, with 15 out of 17 repeats carrying an Asn residue here. Recent structural work showed that consecutive Asn residues present in each repeat at the same position stabilise the LRR architecture by a highly conserved hydrogen bonding pattern from repeat to repeat. This has been termed the asparagine ladder and is considered as the structural backbone of LRR domains[28]. The p.N94S variant, found as a de novo variant in two independent cases, is poised to disrupt this backbone, at least for the first two LRRs in Densin-180. The variant p.C106Y (de novo; pedigree M) may cause spatial conflicts in the densely packed repeats, due to introduction of the large tyrosine residue. p.A193P (inherited from a healthy father; pedigree N) is likely to locally alter LRR secondary structure. p.L221M and p.L296P (both de novo variants; pedigrees O and P) again alter highly conserved Leu residues in repeats 7 and 11. In particular, L296P is likely to disrupt the folding of the LRR region due to the steric constraints introduced by a

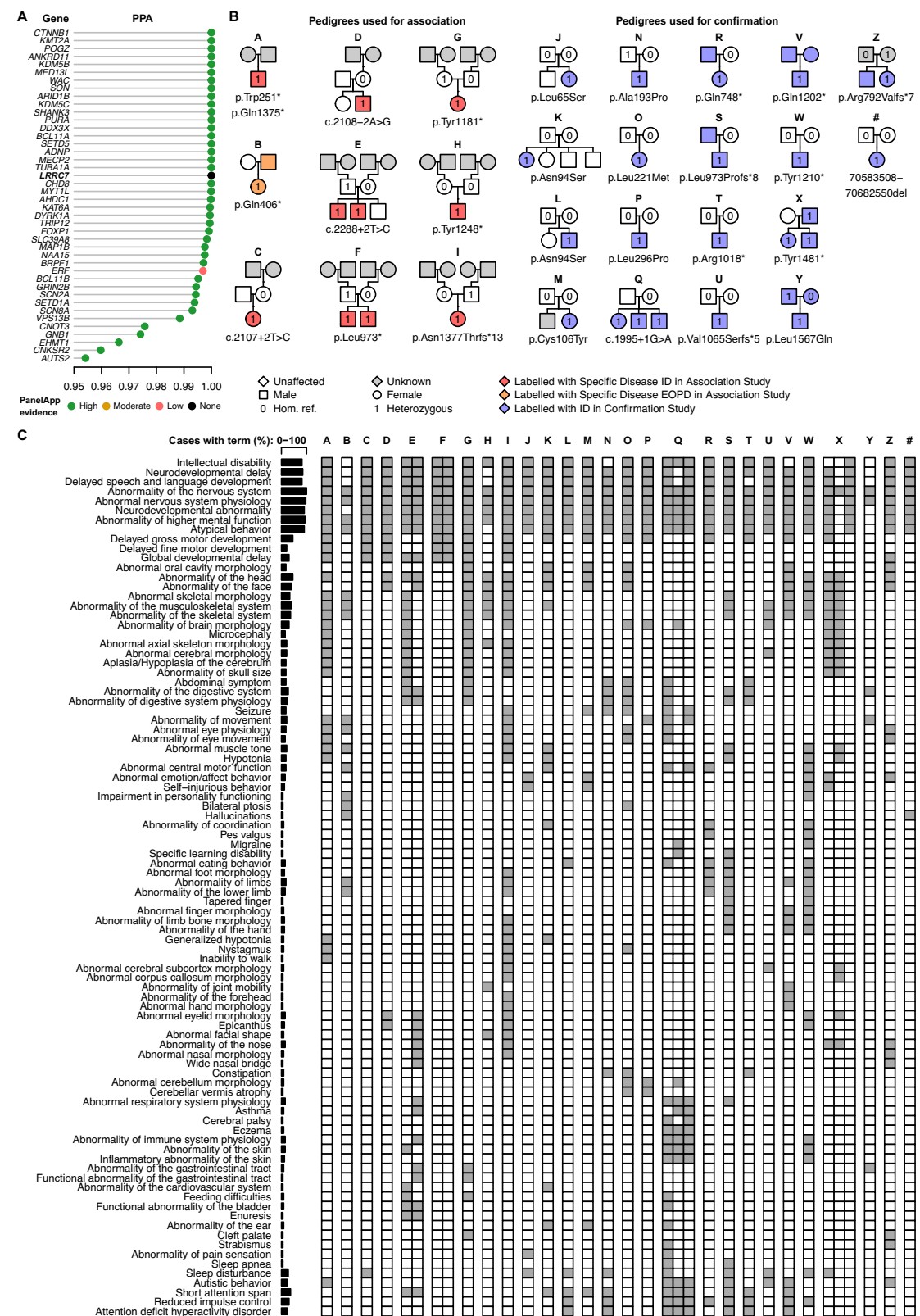

proline residue. Finally, the p.L1567Q variant (pedigree Y, case 20; inherited from father) alters a part of the hydrophobic core of the C-terminal PDZ domain of Densin-180.

### Functional analysis of missense variants in *C. elegans*

*C. elegans* has a single LAP gene, *let-413*, which is homologous to the four human LAP genes[2]. *let-413* functions in apical/basal epithelial cell polarity and in the nervous system[2,29]. Thus, it can be considered as functionally ancestral to its four mammalian paralogs. Out of Densin-180 residues altered in patients, N94, L221 and L296 are conserved in the let-413 protein and Leu65 is replaced by Ile. Cys106 (changed to Leu) and Ala193 (to Ser) are not conserved in let-413. We knocked the p.L221M and p.L296P variants into the corresponding let-413 residue (L173M and L248P), which reside in the homologous LRR7 and LRR11

**Fig. 1 | Association of variants in *LRRC7* with a neurodevelopmental disorder.** **A** Bevimed posterior probability of association (PPAs) > 0.95 with the Intellectual Disability (ID) disease class in the 100KGP. The associations are coloured by their level of supporting evidence in the PanelApp database of gene panels[24] [accessed November 2022] (green: high, red: low, black: absent). **B** Left: Pedigrees A-I of 100KGP participants with probable loss-of-function (pLoF) variants in *LRRC7*. Individuals affected by ID according to 100KGP data are indicated in red; pedigree B stands out as two family members, shown in orange, are affected and assigned to the 'Early onset and familial Parkinson's disease' (EOPD) 100KGP disease class. Right: Pedigrees J-# of the Genematcher/confirmation group of patients; affected individuals are indicated in blue. 1 indicates individuals heterozygous for a truncating variant, 0 indicates individuals homozygous for the allele in the reference genome. Amino acid positions refer to RefSeq entry NM_001370785.2. **C** Grid showing the human phenotype ontology (HPO) terms assigned to the phenotyped members of the nine pedigrees. The columns are grouped by pedigree. Only terms in at least two individuals are shown, and redundant rows have been removed.

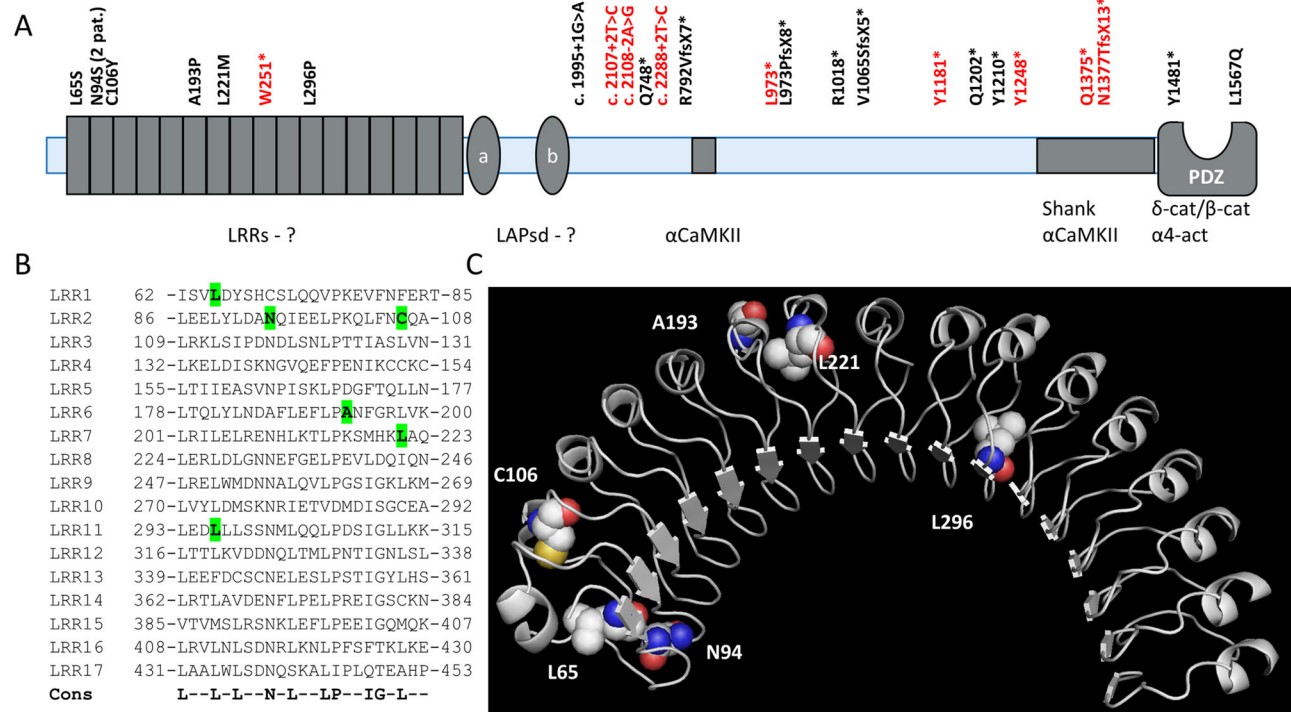

**Fig. 2 | Variants in *LRRC7* associated with a neurodevelopmental disorder.** **A** Domain structure of the Densin-180 protein encoded by *LRRC7*. Positions of missense, nonsense and frameshift variants are indicated, based on database entry NM_001370785.2. Variants from the 100KGP are listed in red, variants obtained through a GeneMatcher contact are shown in black. Known interaction partners of individual domains are indicated. No interaction partners have been published for the leucine rich repeat (LRR) and LAP specific domains (LAPsd) a and b (indicated by?). **B** Sequence alignment of the 17 LRRs of Densin-180, together with a consensus sequence. **C** 3D model of the Densin-180 LRR region, provided by the AlphaFold server. Residues altered by patient variants in *LRRC7* are highlighted in (**B**) and indicated by space filling spheres in (**C**). Note that L65 and L296, as well as C106 and L221, are at equivalent positions in their respective repeats.

repeats, respectively. *let-413* codes for a basolateral protein involved in apical junction assembly and maintenance of cell polarity in embryonic epithelial cells[30]. *let-413* is required for embryonic elongation, the actin dependent process where the coma stage embryo, essentially a ball of cells, undergoes morphogenesis to form a tube-shaped worm and where loss of function variants result in a 1.5-fold stage embryonic arrest, at the beginning of elongation[2]. *let-413* variant [L173M] and control [L173L] edited homozygous worms were indistinguishable from wild type (WT) for viability, growth to adulthood, brood size and behaviours crawling and thrashing. In contrast, the *let-413* variant [L248P], but not the control [L248L], edited homozygous worms are embryonic lethal. Examination of *let-413*[L248P] and *let-413*[Del] homozygotes by Nomarski microscopy demonstrated an embryonic arrest at the 1.5-fold stage, similar to that observed previously in *let-413* premature stop mutants[2]. To examine potential defects in apicobasal polarity and epithelial integrity, we employed the adherens junction marker AJM-1::GFP[31]. In *let-413*[L248L] control edit and *let-413*[Del]/ *tmC3* heterozygous embryos, AJM-1::GFP displays a continuous rectilinear pattern around epidermal cells (Fig. 3), as observed in WT. In contrast, *let-413*[L248P] variant edit and *let-413*[Del] homozygous embryos AJM-1::GFP staining is discontinuous and in many places absent in epithelial cells (Fig. 3), as previously observed in *let-413* premature stop mutants[2]. These results indicate that *let-413*[L248P] is damaging, displaying a strong hypomorphic, possibly null phenotype. By extension, the *LRRC7* variant p.L296P is likely damaging and is a strong hypomorph.

### Truncation of Densin-180 disrupts binding to αCaMKII and δ-catenin

Most nonsense, frameshift and splice site variants detected here are expected to lead to early stop codons, which is likely to be associated with *nonsense mediated decay* (NMD) of the mRNAs after the pioneering round of translation. Nevertheless, we tested for two nonsense variants (p.R1018*, case 13 and p.Y1210*, case 16) whether binding to C-terminal interaction partners is altered. N-terminally GFP-tagged Densin-180 constructs carrying these variants were coexpressed with HA-tagged δ-catenin, and T7-tagged αCaMKII, followed by immunoprecipitation of GFP-tagged Densin-180. As expected, we observed for both interaction partners that the interaction was mostly lost upon these C-terminal deletions (Supplementary Fig. 1). Weak but consistent

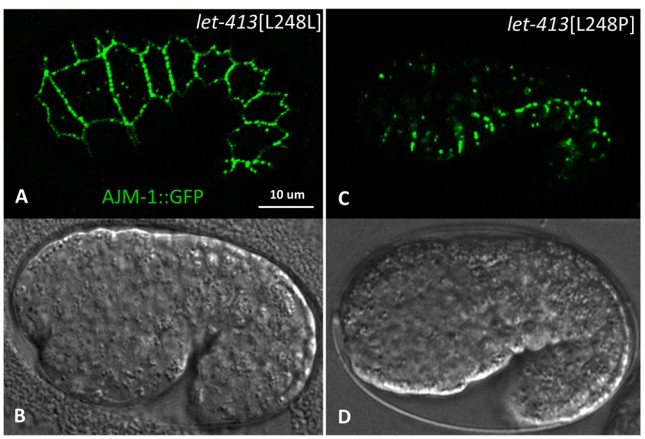

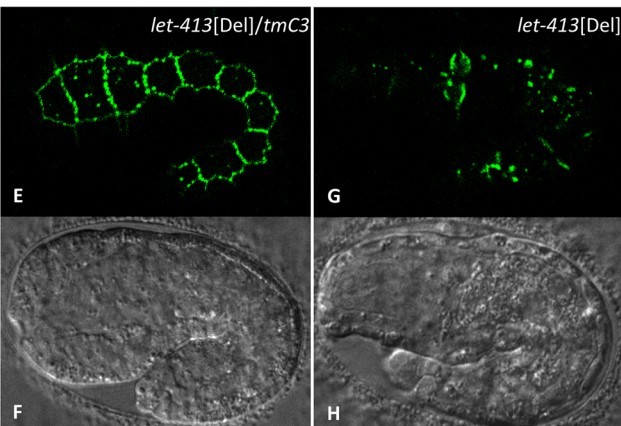

**Fig. 3 | *let-413*[L248P] embryos display an epidermal integrity defect.**
**A**, **C**, **E**, **G** Single external focal plane images of AJM-1::GFP fluorescence. AJM-1 is an adherens junction component that is located at the apicolateral surface of epithelial cells in the (**C**). elegans embryo. **B**, **D**, **F**, **H** Nomarski light microscopy images of the corresponding fluorescent image above. *let-413*[L248L], *let-413*[L248P] homozygous and *let-413*[Del]/*tmC3* heterozygous embryos were obtained 4.5 h after a 1 h egg lay at 25 °C, while the *let-413*[Del] homozygous embryo was 5.5 h after

egg lay. Note WT continuous AJM-1::GFP staining surrounding *let-413*[L248L] epidermal cells (**A**) and *let-413*[Del]/*tmC3* epidermal cells at a slightly higher focal plane (**E**), while *let-413*[L248P] and *let-413*[Del] show discontinuous staining, indicative of disruption of embryonic epithelial cell integrity. Older arrested embryos can show vacuolation and rupture. Scale bar: 10 μm for all images. For microscopic analysis, three separate experiments were performed, with about 50 embryos scored in each experiment with similar results.

binding is maintained for αCaMKII for both variants (Y1210* and R1018*; see quantification). This is in agreement with the presence of a second binding site at amino acids 842–856 of Densin-180, which requires activation of αCaMKII by phosphorylation of Thr286[11,32].

### A missense variant in the PDZ domain of Densin-180 blocks binding to δ-catenin in 293T cells and interferes with postsynaptic localisation of overexpressed Densin-180 in neurons

We analysed the effect of the variants on the molecular interactions of Densin-180, and on the proper postsynaptic targeting in cultured hippocampal neurons. For the PDZ domain variant p.L1567Q, it is known that this domain binds to C-terminal motifs of β- and δ-catenin as well as actinin[7,9,10]. All three proteins carry canonical type I PDZ ligands at their C-termini (consensus: X-S/T-X-Φ-COOH; where X is any amino acid, and the C-terminal Φ is a large and hydrophobic residue). By coexpressing GFP-tagged Densin-180 with β- and δ-catenin in 293T cells, followed by immunoprecipitation and Western Blot analysis, we show that binding is indeed mediated by the C-termini of β- and δ-catenin (Supplementary Fig. 2), and that the *LRRC7* missense variant p.L1567Q found in pedigree Y significantly reduces binding (Fig. 4A, B).

### Variants in Densin-180 differentially affect postsynaptic targeting

Wildtype and missense variants were expressed in primary cultured hippocampal neurons, with the goal to assess possible effects of the mutants on neuronal morphology, synapse formation and on synaptic targeting of the protein (Fig. 4C–F; also Figs. 5, 6). We had previously observed that strong overexpression of Densin-180 in hippocampal neurons, using a CMV promoter-based vector, leads to highly branched dendrites, while the expressed protein was not found in postsynaptic clusters[8]. Here we used an EF1α promoter-based vector to achieve a more moderate expression of Densin-180-GFP in hippocampal neurons. In these experiments, the WT protein was localised to spine-associated dendritic clusters. No significant changes in neuronal morphology (measured as the number of primary dendrites) were observed (Fig. 4C, D). In higher magnification pictures, WT Densin-180 clusters stained positive for the postsynaptic marker (and Densin-180 interacting protein) Shank3, consistent with the known postsynaptic localisation of Densin-180[5,10]. Analysis of the L1567Q variant in the PDZ domain also showed clustering along dendrites; however, we observed that a

reduced portion of these clusters were positive for the postsynaptic marker Shank3 (Fig. 4C, E, F). Thus, the L1567Q variant alters PDZ domain mediated interactions (e.g. with δ-catenin in 293T cells), and this appears to be relevant for postsynaptic targeting of Densin-180.

LRR domain missense variants of Densin-180 were also expressed in hippocampal neurons (Fig. 5 and Supplementary Fig. 3). Again, no effect on the number of primary dendrites was observed; however, Sholl analysis showed reduced branching of dendrites for some of the LRR variants, which became significant for L65S, N94S and L296P variants at some distance points of the measurement (Supplementary Fig. 4). In high magnification pictures of neurons costained for the postsynaptic marker Shank3, we observed that WT and variants L65S, C106Y, A193P and L221M were properly targeted to postsynaptic sites, with L65S actually showing improved postsynaptic targeting when compared to the WT. For the N94S variant, we observed a slight reduction in the formation of postsynaptic (Shank3 positive) clusters of Densin-180 (Fig. 6). In addition, the L296P variant completely failed to reach postsynaptic sites. These data were further confirmed by colocalization analysis of the expressed GFP-tagged Densin-180 with the endogenous Shank3 (Supplementary Fig. 5). In addition, we measured the ratio of fluorescence signal intensity between spine and shaft; here, all variants tested showed spine enrichment, with the exception of the L296P variant which exhibited a diffuse localisation of GFP-fluorescence along the dendrite (Fig. 6). It should be noted that this variant was always expressed at much weaker levels and it was difficult to actually find cells which did express this variant. This may suggest potential folding problems, which were also supported in further experiments in other cellular systems (see below).

None of the variants tested here interfered with the formation of Shank3 clusters, suggesting that synapse formation itself is not altered in the presence of variant or WT Densin-180 (Supplementary Fig. 6A). In addition, we analysed for one variant (L65S) the density of vGlut/Shank3 positive clusters on dendrites (i.e. true synaptic contacts labelled by a pre- and postsynaptic marker), and again did not observe any difference between WT and variant Densin-180 (Supplementary Fig. 6B–D).

### LRR domain variants interfere with binding to novel interaction partners of Densin-180 in 293T cells

From the experiments shown above, we conclude that most missense variants do not interfere with postsynaptic targeting of Densin-180.

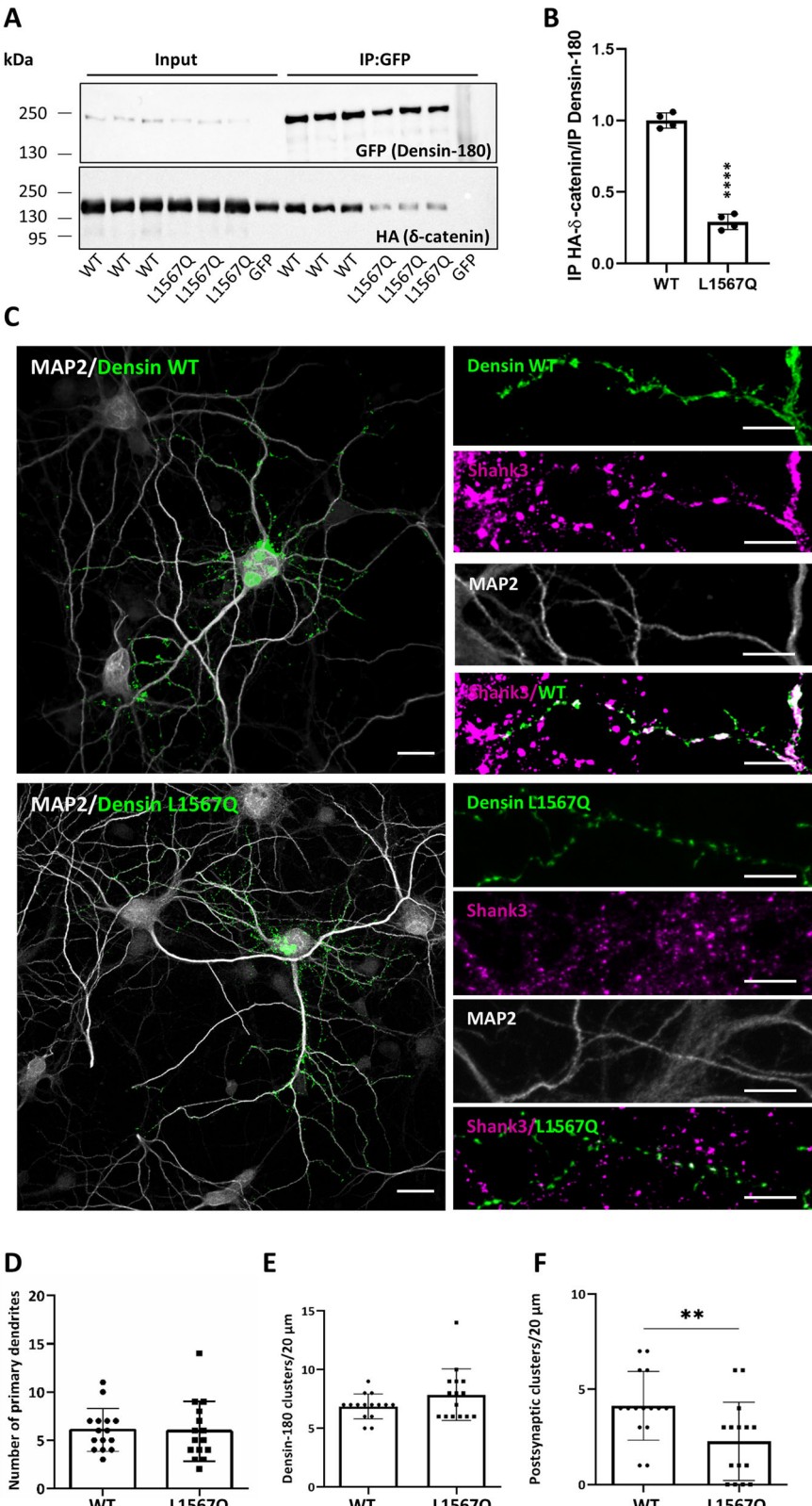

This leads to the question whether any interactions of Densin-180 at the postsynapse may be altered by the LRR variants. First, we tested known interactions in the non-LRR part of Densin-180. As expected, binding to the αCaMKII, Shank3 and δ-catenin was not altered by the LRR domain variants (Supplementary Fig. 7). To our knowledge, no interaction partners for the LRR of Densin-180 have been identified before. Therefore, we cast a rather wide net for possible interactors of

the LRR domain. Through a yeast two hybrid screen, we discovered a possible interaction with C-terminal parts of the microtubule actin crosslinking factor 1 (MACF1) and of β-spectrin. Out of these, β-spectrin interacted only weakly in a coexpression/co-IP experiment in 293T cells, whereas the C-terminal fragment of MACF1 showed a robust interaction (Supplementary Fig. 8). By proximity (BioID) labelling in transfected 293T cells, we identified Nectin-2, Afadin, Erbin and

**Fig. 4 | A variant in the PDZ domain interferes with binding to δ-catenin and targeting to postsynaptic clusters. A** 293T cells coexpressing GFP-tagged Densin-180 (full-length; WT or L1567Q mutant), or GFP alone, together with an HA-tagged δ-catenin were lysed (Input samples). GFP-tagged proteins were immunoprecipitated (IP) using the GFP-trap matrix. Input and precipitate samples were analysed by Western blotting using the epitope tag antibodies, as indicated. **B** Quantification of data from four independent transfections. Data are presented as the ratio of HA-δ-catenin signals in IP samples, divided by GFP-Densin signals, also in IP samples. Data are shown as mean values ± SD. ****, significantly different; p < 0.0001; unpaired, two sided Student's *t* test. **C** Hippocampal neurons were transfected with plasmids coding for Densin-180 WT or the L1567Q mutant, and stained for MAP2,

Shank3 and the expressed protein. Cells were analysed by confocal microscopy. Left panel shows overview images of neurons (scale bar 20 μm) and right panel shows magnified dendritic segments (scale bar 10 μm). Three independent preparations were transfected and neurons were analysed, with similar results. Quantitative analysis showed that the number of primary dendrites (**D**), and the number of dendritic Densin-180 clusters (**E**), is not affected by the L1567Q variant. However, the number of postsynaptic clusters of Densin-180, defined by colocalization with Shank3, is reduced (**F**). Quantitative analysis was performed on 15 neurons (**D**) and on 45 dendrites of 15 neurons (**E, F**) obtained in three independent preparations. **, significantly different from WT; p = 0.009 paired, two sided t-Test. Data are shown as mean values ± SD. Source data are provided as a Source Data file.

the RNA helicase DDX3X as potential partners of the Densin-180 LRR domain; interactions with DDX3X and the Densin-180 homologue Erbin (and Densin-180 itself) were reproducible in co-IP experiments (Fig. 7A). Furthermore, a literature search showed that LRRs of the related protein Scribble bind to the band 4.1 like protein 1 (EPB41L1) and the catalytic subunits of protein phosphatase 1 (PP1), such as PP1α[33]. Strong binding was confirmed for PP1α in coimmunoprecipitation experiments from 293T cells (Fig. 7B). MACF1, Erbin and PP1α were chosen for further analysis.

Patient derived missense variants in the LRR were introduced into a construct coding for GFP-tagged full-length Densin-180, and into a construct coding for LRR-GFP only. These constructs were transfected into 293T cells together with constructs coding for the identified interaction partners (Fig. 8). Upon cell lysis and immunoprecipitation of GFP-tagged Densin-180 variants, most variants in the LRR strongly and significantly interfered with binding to MACF1, with the exception of the A193P and L221M variants. Similar results were obtained when the interaction with Erbin was analysed. Here, again most tested variants showed a significant reduction in binding to Erbin (including the A193P variant), while L221M failed to show an effect here. Finally, strong binding of the LRRs to PP1α was reduced significantly by L65S, N94S, C106Y and L296P variants, while A193P and L221M did not have any effect. For L65S and N94S variants, we also verified that binding to PP1α is lost in the context of the full-length Densin-180 protein (Supplementary Fig. 9). In summary, four variants at key positions of the LRR consensus sequence (L65, N94, C106 and L296) severely disrupt binding to all partners tested. A193P showed a reduction in binding for only one of the partners, while L221M did not show any statistically significant effects. Consistently, the A193P variant has been inherited from a healthy father (Supplemental Data 1). The L221M variant did not affect the ability of let-413 to support the establishment of apicobasal polarity in the *C. elegans* embryo (Fig. 3). Thus, A193P appears to be a rather mild variant, and L221M, despite affecting a highly conserved residue in the LRR domain, should be reconsidered as a VUS (variant of unknown significance).

Interactions with PP1α via the LRR, and αCaMKII via the central and C-terminal binding sites, suggested that Densin-180 may function as a scaffold for a kinase/phosphatase complex at the synapse. Indeed, we had observed in coexpression/coimmunoprecipitation experiments that both proteins can bind to Densin-180 (Fig. 7, Fig. S7). We further investigated this by expressing an mRFP-tagged mutant of the αCaMKII (T286E), which mimics the phosphorylated, active form of the kinase. Upon immunoprecipitation of tagged αCaMKII (both WT and T286E mutant), we observed some coprecipitation of PP1α; this was significantly increased when GFP-tagged Densin-180 was coexpressed (and coimmunoprecipitated; Supplementary Fig. 10). These data are compatible with a model where Densin-180 may scaffold a αCaMKII/Densin-180/PP1α triple complex. It is unclear whether endogenous proteins in 293T cells also support such a complex, but Erbin or Scribble are possible candidates.

Densin-180 appears to be almost exclusively localised at postsynaptic sites. Biochemical fractionation of the mouse cortex supports

this view, as Densin-180 is strongly enriched in the PSD fraction. Our analysis shows that PP1α is also strongly enriched in the PSD fraction, together with Densin-180 (Supplementary Fig. 11A, B). This is consistent with immunocytochemical staining of cultured hippocampal neurons, which shows colocalization of expressed Densin-180 WT and PP1 in presumably postsynaptic (Shank3 positive) clusters along dendrites (Supplementary Fig. 11C, D).

Both the biochemical as well as the microscopic analysis show that PP1 has a much wider distribution than Densin-180, as it is known to operate in several distinct protein complexes in neurons. Taken together, our data suggest that Densin-180 may be needed to target PP1α to specific substrates at postsynaptic sites, and that this scaffolding function may be disrupted by missense variants in the LRR. This is likely to contribute to intellectual disability of patients identified here.

A summary of the results of all functional analyses performed on missense variants of *LRRC7* is provided in Table 1.

## Discussion

We show here that rare variants in *LRRC7*, coding for Densin-180, lead to a neurodevelopmental disorder in humans. The recent analysis of the 100KGP revealed a strong association of truncating variants in *LRRC7* with a neurological disorder[19]. Here, we combined the extraction of data from the 100KGP with a large collection of patients derived from individual genetic laboratories which were linked together through the GeneMatcher tool[25]. A previous report, which identified one case of ID associated with a truncating *LRRC7* variant, has been included here as case #16[20]. In total, we identify 33 individuals with variants in *LRRC7* affected by a neurological disorder.

The core phenotype of these individuals consists of ID and delayed language development, in addition to ADHD, autism and seizures in several cases. In this respect, *LRRC7* is similar to several other genes coding for synaptic proteins[34,35]. Furthermore, aggressive or impulsive behaviour is prevalent in many cases reported here (six males, three females out of 22 cases in the replication cohort). Genome wide association studies pointed to a role of *LRRC7* in childhood behavioural disorders, including ADHD and aggressive behaviour[36]. Interestingly, two independent studies on *Lrrc7* deficient mice have observed a strong aggression phenotype in homozygous, but also in male heterozygous ko mice[13,14].

As a third remarkable phenotype, we noted hyperphagia associated with obesity, starting between 4 and 10 years of age. Case descriptions are reminiscent of cases of Prader-Willi syndrome with patients foraging for food regardless of its source (see case 12/pedigree S). Hyperphagia was observed in 4 out of 14 cases with truncating variants described in Supplemental Data 1; however, several of the other patients were still too young to observe this phenotype and a follow up will be needed later in life.

More than half of the patients described here carry a truncating, presumably loss-of-function variant. We expect the respective transcripts to be degraded by the NMD pathway due to splice site, frame shift and early nonsense variants. This suggests haploinsufficiency for

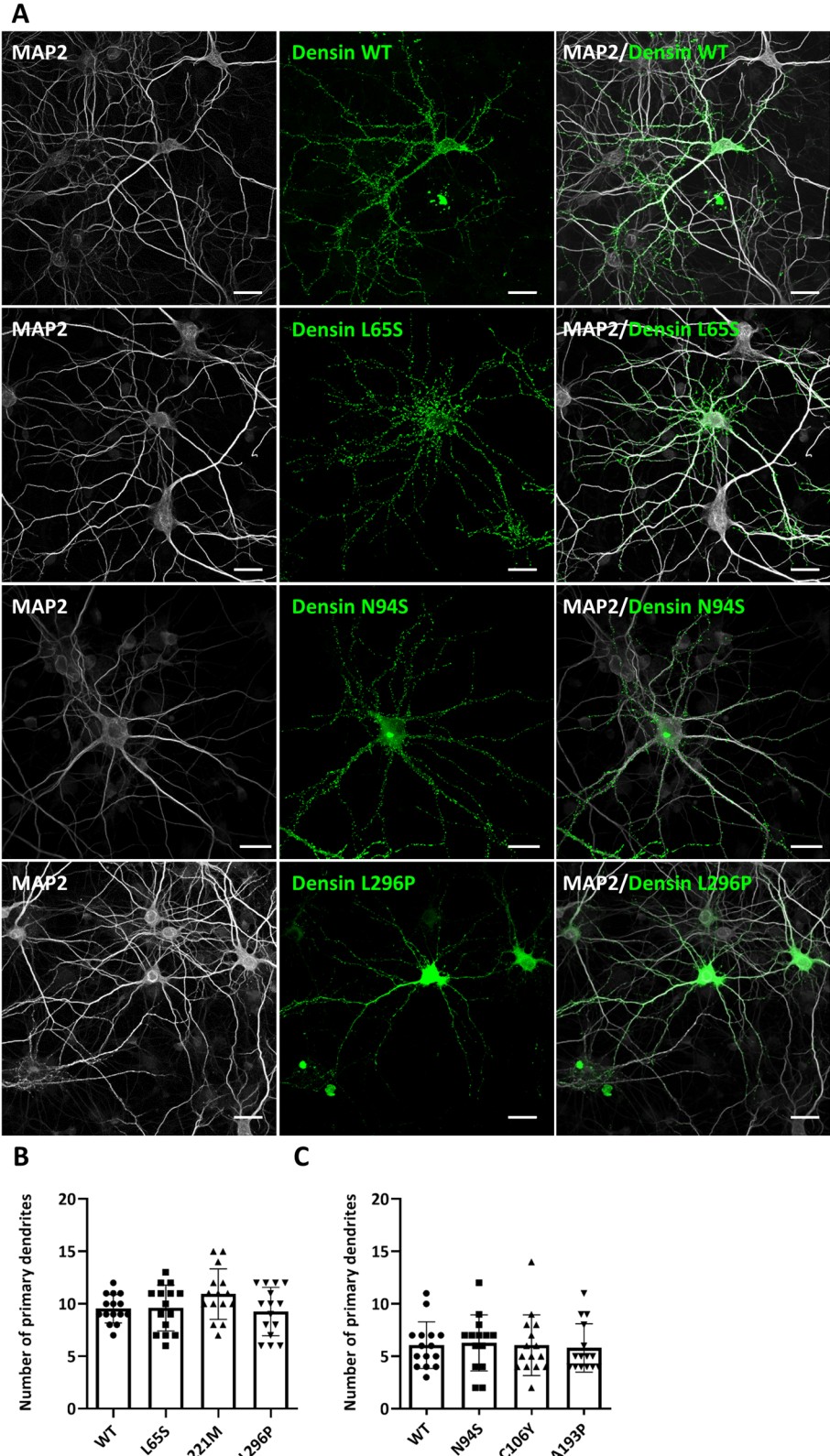

**Fig. 5 | Expression of Densin-180 LRR domain variants in cultured neurons.**
**A** Primary cultured hippocampal neurons were transfected with plasmids coding for GFP-tagged Densin-180 (WT and variants) at day in vitro 7 (DIV7). After differentiation for a total of 14 days (DIV14), cells were fixed, stained for MAP2 and visualised by confocal microscopy (scale bar: 20 μm). Number of primary dendrites emerging from cell somata for variants shown in this Figure (**B**) and for variants shown in Supplementary Fig. 3 (**C**). Differences are non-significant upon analysis of n = 15 neurons from three independent transfections; one-way ANOVA, followed by Dunnett's multiple comparisons test. Data are shown as mean values ± SD. Source data are provided as a Source Data file.

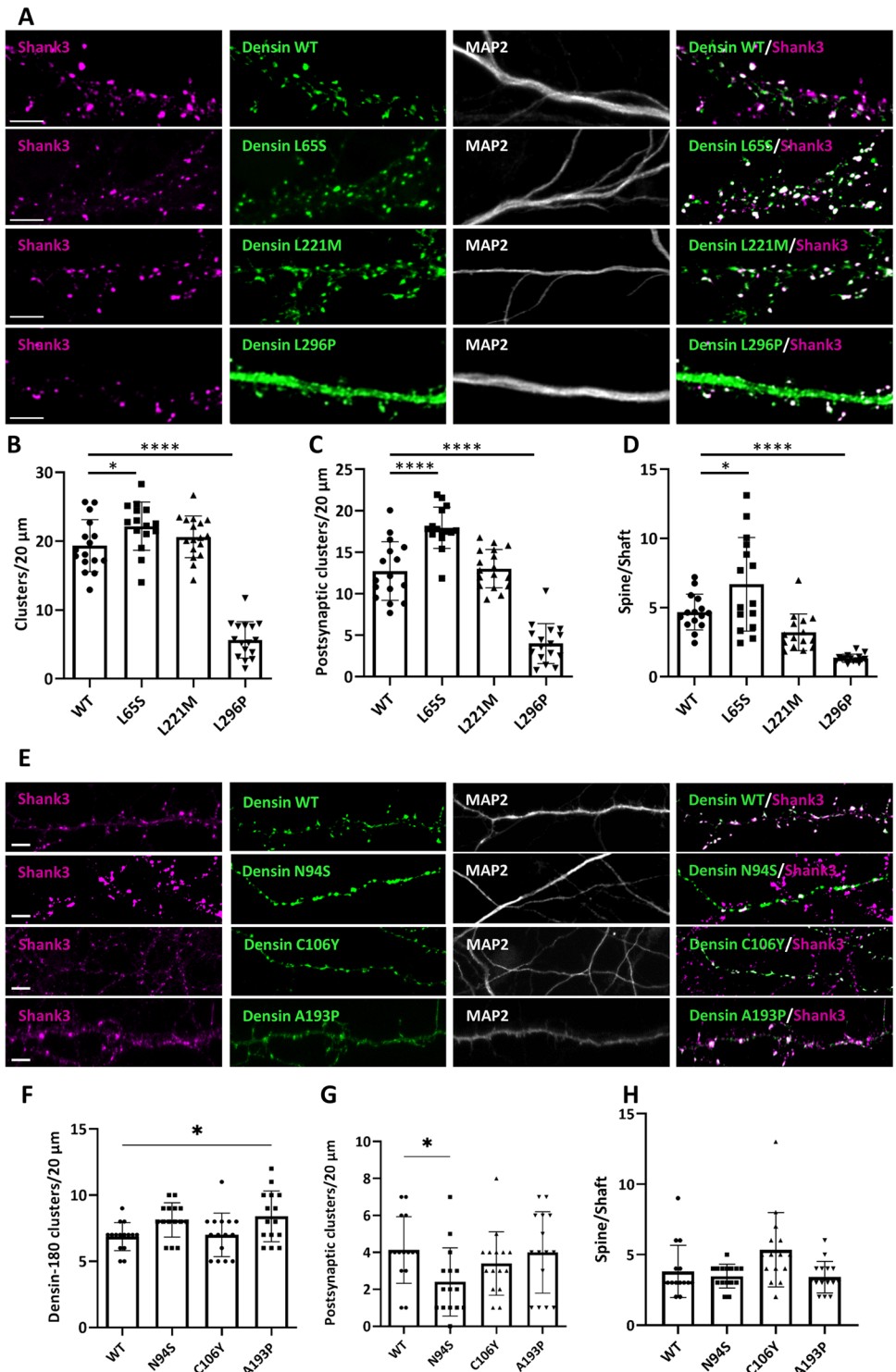

**Fig. 6 | Variants N94S and L296P affect targeting of Densin-180 to postsynaptic sites. A** High magnification micrographs of hippocampal neurons expressing GFP-Densin-180 WT and variants L65S, L221M and L296P (scale bar: 5 μm). Neurons were costained for the postsynaptic marker Shank3, and the somatodendritic marker MAP. For quantitative analysis, we counted the number of clusters per length of dendrite (**B**; p-values 0.0114 for L65S; 0.3448 for L221M; < 0.0001 for L296P), the number of Shank3-positive GFP-Densin clusters (**C**; p-values < 0.0001 for L65S; 0.9484 for L221M; < 0.0001 for L296P), and the spine-shaft ratio of GFP-fluorescence intensities (**D**; p-values < 0.0165 for L65S; 0.1007 for L221M; < 0.0001 for L296P). Data are shown as mean values ± SD. **E**–**H** The analysis was repeated for N94S, C106Y and A193P variants (scale bar: 5 μm). p-values for differences to WT are for (**F**): 0.0675 (N94S); 0.9903 (C106Y); 0.0208 (A193P); for (**G**): 0.0414 (N94S); 0.5894 (C106Y); 0.9950 (A193P); for (**H**): 0.9168 (N94S); 0.0531 (C106Y); 0.8678 (A193P). Data are shown as mean values ± SD. *,**,***,**** indicate statistically significant differences from WT, p < 0.05, 0.01, 0.001, 0.0001, respectively. Analysis was performed on 45 dendrites (**B**, **C**, **E**, **F**) and 150 dendritic clusters (**D**, **G**) of 15 neurons from three independent transfections. One-way ANOVA, followed by Dunnett's multiple comparisons test. Source data are provided as a Source Data file.

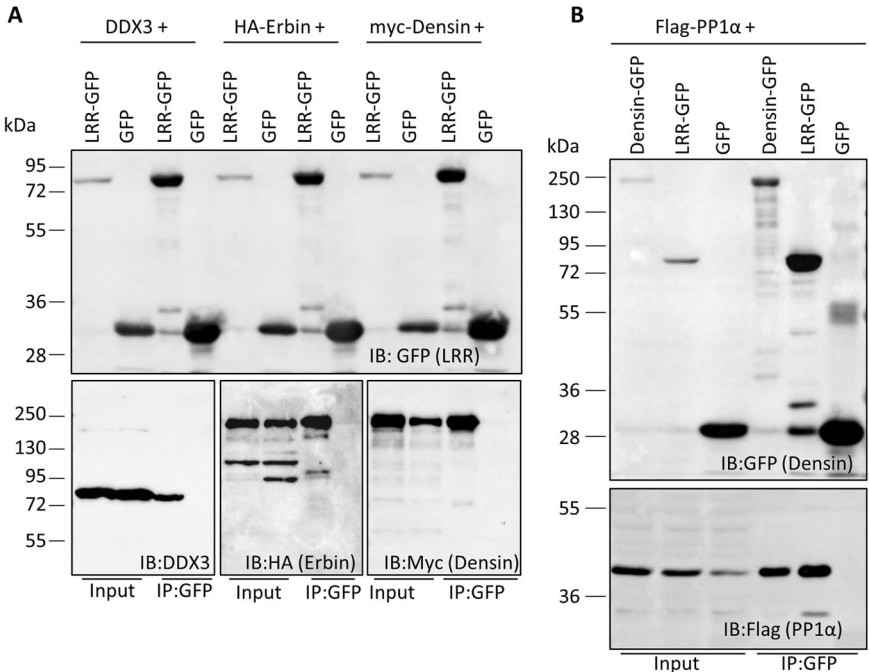

**Fig. 7 | Identification of novel interaction partners of the LRR domain of Densin-180. A** 293T cells were transfected with plasmids coding for GFP-tagged LRR of Densin-180, or GFP alone, in combination with DDX3, HA-tagged Erbin, or myc-tagged full-length Densin-180. After cell lysis, GFP-tagged proteins were immunoprecipitated (IP) using the GFP-trap matrix. Input and IP samples were analysed by Western Blot (immunoblot, IB) using the antibodies indicated.

**B** 293T cells were transfected with plasmids coding for GFP-tagged Densin-180 (full length), GFP-tagged LRR domain, or GFP alone in combination with Flag-tagged PP1α. GFP-tagged proteins were immunoprecipitated as in (**A**) and analysed by Western blotting. Experiments were repeated three times, with similar results. Source data are provided as a Source Data file.

the encoded Densin-180 protein as a major pathomechanism, consistent with human population data from gnomAD (pLI = 1, o/e = 0.13, gnomAD v4.1.0). We noted that in several cases, these variants were inherited from healthy parents, indicating incomplete penetrance of the phenotype. So far, it is unclear if the condition always manifests.

Currently, we did not observe that inherited variants lead to a less severe phenotype than de novo variants. Phenotypic heterogeneity is evident in cases who share the same variant, such as mother and daughter in pedigree Z, where the daughter has a rather typical phenotype, whereas the mother is affected by a neuropsychiatric disorder but no ID. Strikingly, all missense variants which disrupt binding to the newly identified interaction partners of the LRR occurred de novo. However, these variants are not associated with a particularly severe phenotype.

Further insight into the functional aspects of Densin-180 required for normal brain function were provided by these missense variants. Only one of those affected a known interaction partner of Densin-180, i.e. δ-catenin, which (similar to β-catenin and α-actinin) interacts with the PDZ domain of Densin-180[7,9,10]. The L1567Q variant diminishes this interaction strongly, and also interferes with the correct synaptic targeting of Densin-180.

As most missense variants affected the LRR domain, we invested considerable efforts into the identification of ligands for this domain, which were not known. We identified Erbin, another LAP protein; this pointed to the possibility of hetero- or homo-dimerisation of LAP proteins, which had been observed before for Scribble[33]. Indeed, LRR-mediated association between LAP proteins was observed, and the Densin-180/Erbin complex was disrupted by five out of six variants found in patients.

As a second putative partner, we identified a C-terminal fragment of MACF1, containing two EF-hands and part of a *growth-associated 2 region* (GAR) domain. Variants in the MACF1 gene have been associated with neurological disorders, in particular lissencephaly[37]. Interaction of

Densin-180 with the C-terminal fragment of MACF1 was again disrupted by missense variants, in this case by four out of five LRR domain variants tested. MACF1 may be responsible for the effects of Densin-180 on neuronal morphology identified by us and others[8,14]. However, we did not investigate this further as MACF1 is an extremely large protein (more than 7000 amino acids).

Finally, we identified PP1α as a strong and reproducible interaction partner of the LRR domain of Densin-180 in coexpression/coimmunoprecipitation experiments from 293T cells, similar to what was described before for Scribble[33]. PP1 isoforms PP1α and PP1γ are highly enriched in dendritic spines and in the postsynaptic density[38]—i.e. at the same subcellular location as Densin-180[5]. Interestingly, recent BioID experiments identified Densin-180 as one of the few proteins, which are present at inhibitory as well as excitatory postsynaptic complexes[39]. PP1α was also enriched in inhibitory postsynaptic complexes[39], suggesting that the Densin-180/PP1α might be present at excitatory as well as inhibitory synapses. Here, we found most Densin-180 colocalizing with Shank3 at excitatory synapses.

Patient variants at key positions of the LRR consensus (L65, N94, C106 and L296) disrupt binding to all partners tested, raising mechanistic questions. The interactions of PP1 with LRR domain proteins have been structurally investigated for SDS22[40] and Shoc2[41–44]. These analyses show that PP1 uses most of the concave side of the LRR (see Fig. 2C) for binding, and that there is even space for a second interactor on this concave surface in the case of Shoc2. Also for Scribble, binding of PP1 and the dimerisation of scribble via the LRR have been attributed to interactions at the concave site[33]. The patient variants in question are likely to alter the packing between individual repeats, thereby changing the curvature of the concave surface in a way, which is incompatible with binding.

PP1 variants are targeted to their substrates via adapter or scaffold proteins, thereby providing specificity in the negative regulation of kinase-based signalling events[45]. This leads to the question: which

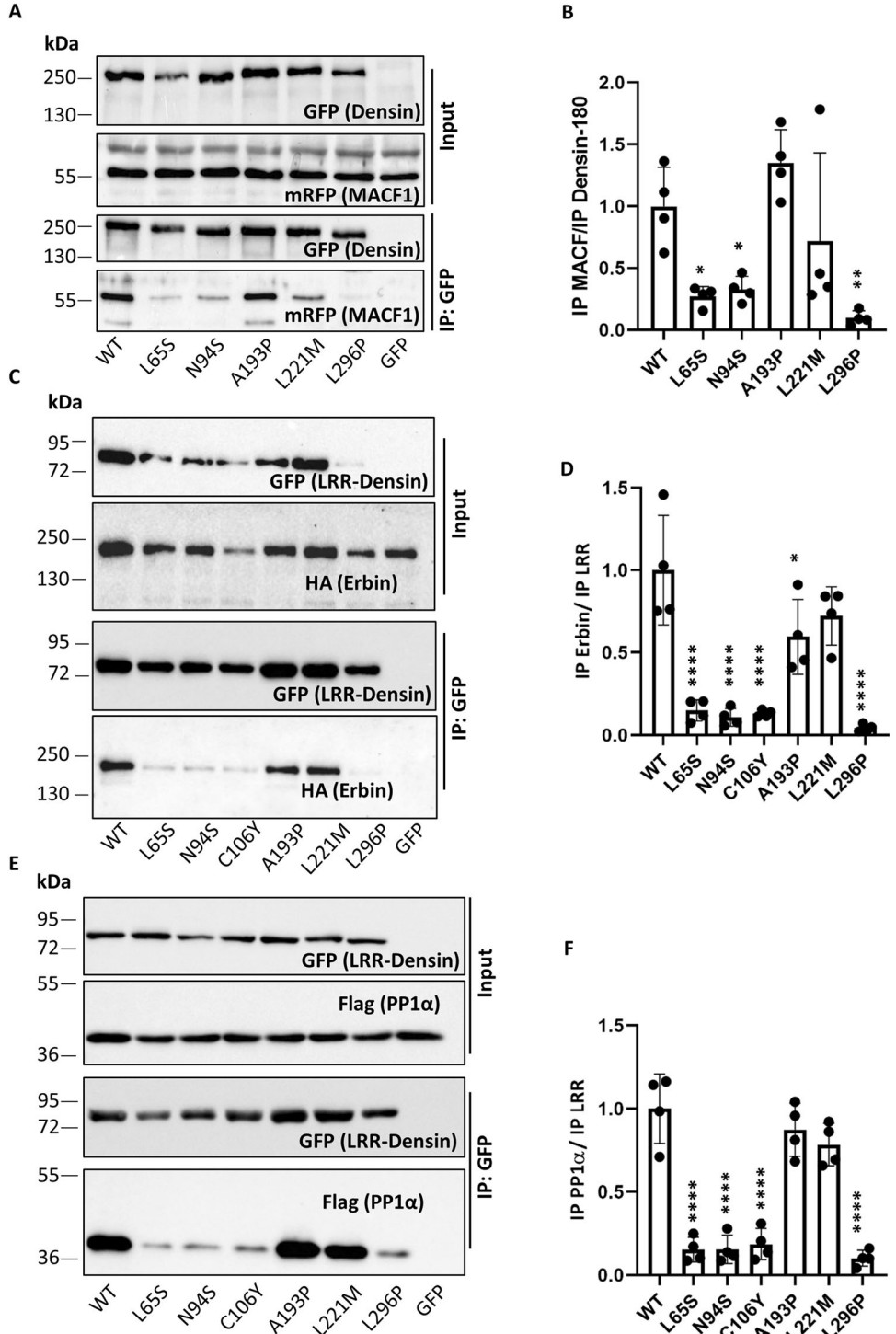

**Fig. 8 | Variants in *LRRC7* disrupt binding to interaction partners of the LRR domains in 293T cells. A** 293T cells coexpressing GFP-tagged Densin-180 (full-length; WT or mutants), or GFP alone, together with an mRFP fusion protein of the C-terminus of MACF1 were lysed (Input samples). GFP-tagged proteins were immunoprecipitated (IP) using the GFP-trap matrix. Input and precipitate samples were analysed by Western blotting using the epitope tag antibodies, as indicated. **B** Quantification of the data shown in (**A**). Data are presented as the ratio of mRFP-MACF1 signals in IP samples, divided by GFP-Densin signals, also in IP samples. p-values for differences to WT are: 0.0298 (L65S); 0.0481 (N94S); 0.4864 (A193P); 0.6646 (L221M) and 0.0064 (L296P). Data are shown as mean values ± SD. **C** 293T cells coexpressing GFP-tagged LRR repeats of Densin-180 (WT or mutant) together with HA-tagged Erbin. Immunoprecipitation and analysis by Western Blot was performed as in (**A**). **D** Quantitative analysis of the data shown in (**C**). Data are

presented as the ratio of HA-Erbin signals in IP samples, divided by GFP-LRR signals, also in IP samples. p-values for differences to WT are: < 0.0001 (L65S); <0.0001 (N94S); <0.0001 (C106Y); 0.0138 (A193P); 0.1276 (L221M) and <0.0001 (L296P). Data are shown as mean values ± SD. **E** Coexpression and coimmunoprecipitation analysis was performed with GFP-tagged LRRs of Densin-180, and Flag-tagged protein phosphatase 1α (PP1α). **F** Quantitative analysis of the data shown in E. Data are presented as the ratio of Flag-PP1α signals in IP samples, divided by GFP-LRR signals, also in IP samples. p-values for differences to WT are: <0.0001 (L65S); <0.0001 (N94S); <0.0001 (C106Y); 0.531 (A193P); 0.1004 (L221M) and <0.0001 (L296P). Data are shown as mean values ± SD. *, ****, indicate significant differences from WT; p < 0.05, 0.0001, respectively. Data were analysed by one-way ANOVA, followed by Dunnett's multiple comparisons test (**B**, **D**, **F**). Data were from four independent transfections. Source data are provided as a Source Data file.

**Table 1 | Summary of effects of *LRRC7* missense variants**

| Variant | L65S | N94S | C106Y | A193P | L221M | L296P | L1567Q |
|---|---|---|---|---|---|---|---|
| Polarity in *C.elegans* | nd | nd | nd | nd | ✕ | ↓ | nd |
| Dendritic clusters | ⇧ | ✕ | ✕ | ⇧ | ✕ | ↓ | ✕ |
| Postsynaptic clusters | ⇧ | ↓ | ✕ | ✕ | ✕ | ↓ | ↓ |
| Shank3 colocalization | ⇧ | ↓ | ✕ | ✕ | ✕ | ↓ | ✕ |
| Spine/shaft ratio | ⇧ | ✕ | ✕ | ✕ | ✕ | ↓ | ✕ |
| Shank3 cluster density | ✕ | ✕ | ✕ | ✕ | ✕ | ✕ | ✕ |
| Primary dendrites | ✕ | ✕ | ✕ | ✕ | ✕ | ✕ | ✕ |
| Branching (Sholl) | ↓ | ↓ | ✖ | ✖ | ✖ | ↓ | ✖ |
| Shank3 binding | ✕ | ✕ | ✕ | ✕ | ✕ | ✕ | ✕ |
| δ-catenin binding | ✕ | ✕ | ✕ | ✕ | ✕ | ✕ | ↓ |
| αCaMKII binding | ✕ | ✕ | ✕ | ✕ | ✕ | ✕ | ✕ |
| MACF1 binding | ↓ | ↓ | nd | ✕ | ✕ | ↓ | nd |
| Erbin binding | ↓ | ↓ | ↓ | ↓ | ✕ | ↓ | nd |
| PP1α binding | ↓ | ↓ | ↓ | ✕ | ✕ | ↓ | nd |

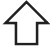

PP1 substrates might be addressed by a Densin-180/PP1 complex? One good candidate is the α-subunit of the Ca²⁺/calmodulin dependent protein kinase II (αCaMKII). Densin-180 has two different binding sites for αCaMKII; there is a central sequence with similarity to canonical CaMKII substrates, (residues 842–856 of human Densin-180) which binds to the active site of αCaMKII. In addition, a C-terminal region of Densin-180 preceding the PDZ domain shows strong binding affinity[10–12,32]. Using deletion constructs based on variants found in case 13 (p.R1018*) and case 16 (p.Y1210*) we confirm that most of the binding to αCaMKII is lost upon deletion of the last 350 amino acids, but some specific binding remains, likely at the central site.

Both αCaMKII and PP1 are targeted to postsynaptic sites by several anchoring proteins, including Shank3 and GluN2B for αCaMKII[46,47] and Neurabin and Spinophilin for PP1[48,49]. Our data show that Densin-180 has the capacity to link both proteins. Within the dense molecular matrix of the PSD, it may be necessary that the enzymatic activities of kinase and phosphatase are physically linked by a scaffold protein to achieve the required speed in signalling. In agreement with this, αCaMKII phosphorylation at Thr286 upon stimulation of NMDA receptors is higher in neurons derived from *Lrrc7* ko mice compared to WT[13]. This is consistent with a model where PP1, anchored to αCaMKII via Densin-180, is responsible for rapid dephosphorylation of αCaMKII during NMDA receptor induced forms of synaptic plasticity. Strikingly, Densin-180, PP1 activity and αCaMKII are all required for establishment of LTD[13,50,51]. As we show here that variants altering the LRR domain of Densin-180 interfere with binding to PP1 in 293T cells, we propose a pathomechanism where disruption of the Densin-180/PP1 complex leads to alter targeting of PP1 to its substrates, thus changing the phosphorylation status of synaptic proteins during synaptic plasticity.

A linkage between ID and deficits in synaptic plasticity (in particular LTD) has been established for other ID syndromes (e.g. fragile X syndrome[52];). Currently, it remains unclear how other phenotypic aspects of patients carrying *LRRC7* variants may be explained. As expression of *LRRC7* is limited to the brain, further studies will need to address central aspects of the control of food intake and control of aggression, for a better understanding of the aetiology of this syndrome.

## Methods

### Ethics statement
Our research complies with all ethical regulations, in particular regarding work with patients (Ethics Committee of the Hamburg Chamber of Physicians: PV 3802); and work with animals (Animal Welfare Committee of the University Medical Center Hamburg-Eppendorf under permission number Org766).

### Analysis of the 100 KGP
We used the Bayesian statistical method BeviMed[53] to obtain PPAs between each of the 19,663 protein-coding genes and the disease class Intellectual Disability[19]. The associations were cross-referenced with the PanelApp database of gene[24] panels to identify their degree of support in the literature[19]. We obtained pedigree data and HPO terms from the rare_diseases_pedigree_ped_member and rare_diseases_participant_phenotype tables of the 100KGP database, respectively. Out of the HPO terms (explicit or implied) present in at least two participants with *LRRC7* pLoF variants, we applied an algorithm to remove redundant terms for display. Redundant terms were defined as those which were present in the same individuals as another term, and were implied by the ontological structure of the HPO.

### Research subjects
Written informed consent for all subjects was obtained in accordance with protocols approved by the respective ethics committees of the institutions involved in this study (approval number by the Ethics

Committee of the Hamburg Chamber of Physicians: PV 3802). Study participants or their legal guardians have affirmed consent for publication of indirect identifiers and clinical information, and have also affirmed consent to publish in an online Journal. Sex and/or gender was not considered in the study design. Sex of participants was determined based on self-report.

### Antibodies
The following primary antibodies were used: mouse anti-GFP (Covance MMS-118P-500, RRID: AB_291290; WB: 1:3000); rat anti-RFP (Chromotek 5F8, WB: 1:1000); rabbit anti-phospho-CamKII (Abcam 32678, WB 1:1000); rabbit anti CamKII (Abcam 52476, WB 1:1000); mouse anti-HA (Sigma-Aldrich H9658, WB 1:1000); mouse anti-cMyc (Sigma-Aldrich C3956, WB 1:1000); mouse anti-T7 tag (Novagen 695522, WB 1:1000); rabbit anti-GFP (Abcam AB6556; ICC: 1:500); chicken anti-MAP2 (Antibodies-Online ABIN361345, ICC: 1:1000); guinea pig anti-Shank3 (Synaptic Systems 162 304; ICC 1:1000); rabbit anti-VGlut (Synaptic Systems; ICC: 1:2000). HRP-labelled goat secondary antibodies were from Jackson ImmunoResearch and used for WB at 1:2500 dilution. For ICC, Alexa 405 goat anti-chicken IgG (abcam; ab175675 at 1:1000 dilution) and Cy3 goat anti-rabbit IgG (Dianova) were used at 1:1000 dilution.

### Expression constructs
Expression constructs coding for rat Lrrc7/Densin-180 (NM_001393674.1, full length, or residues 1-469, encompassing the LRR region only) in pEGFP-N2, leading to a C-terminal fusion with EGFP, have been described[8]. For expression in neurons, a cDNA fragment coding for the Densin-180-EGFP fusion protein was subcloned into the pHAGE vector[18,54] which uses an EF1α promoter instead of a CMV promoter (obtained from Alex Shcheglovitov, University of Utah). Patient variants were introduced by site directed mutagenesis using the Quik-Change kit from Agilent, following the manufacturer's instructions; for each mutant, two complementary oligonucleotides (Sigma-Aldrich, Steinheim, Germany) carrying the desired variant were used in the reaction. Oligonucleotides are listed in Supplementary Table 3. All constructs were verified by Sanger sequencing along the *Lrrc7* coding sequence. All DNA constructs used in this study are available from the corresponding author upon request.

For the BioID screen, cDNA coding for a modified Biotin ligase (BioID2, described in ref. 55) was obtained from K. Roux (Sioux Falls, South Dakota) through Addgene (#92308). The EGFP coding sequence in the pEGFP-N1 vector (Clontech) was replaced with the BioID2 coding sequence, generating pBioID-N1. cDNAs coding for full length Densin-180, or the Densin-180 LRR region were cloned into pBioID-N1, leading to expression of Densin-180 or the LRR domain fused to the biotin ligase at the C-terminus. For yeast two hybrid screening, cDNA coding for the LRR domain was cloned into the pGBKT7 vector (Clontech).

Flag-neo-PP1-Alpha was a gift from Steven Balk, provided through Addgene (# 171267); expression vectors for δ-catenin and Shank3 have been described[56]; cDNAs coding for targets from the yeast two hybrid screen were subcloned into pEGFPC1, and also pmRFP-C1; a vector coding for HA-tagged Erbin was obtained from Jean-Paul Borg (Inserm, Marseille, France). Full length αCaMKII in a modified pcDNA3 vector coding for an N-terminal T7-tag has been described before[8]. The cDNA was subcloned into pmRFP-C1, and a T286E mutant was generated by site-directed mutagenesis.

### Screens for interaction partners of the Densin-180 LRR region
For BioID screening, 293T cells were transfected with constructs coding for Densin-180-BioID2 fusions, or BioID2 control. 24 h after transfection, 50 µM Biotin was added to the media and the cells were cultivated for another 24 h. Cells were lysed with RIPA buffer and lysates were cleared by centrifugation (20,000 × *g*, 15 min).

Biotinylated proteins were purified using NeutrAvidin Agarose resin (Thermo Scientific), followed by five washes with RIPA buffer. Purified proteins were released by boiling in 1x Laemmli sample buffer, and electrophoresed into the first 5 mm of an 8 % SDS separating gel. This upper section of the gel was cut out with a scalpel, and processed for tryptic digestion and identification of proteins. For this, in-gel reduction and alkylation was achieved with 10 mM dithiothreitol, followed by 55 mM iodacetamide in 100 mM $NH_4HCO_3$. Proteins were then digested with a trypsin (8 ng/μL sequencing-grade trypsin, dissolved in 50 mM $NH_4HCO_3$ containing 10% actonitrile) and incubating the mixture at 37 °C overnight. Tryptic peptides were extracted with 2% formic acid, 80% acetonitrile. After evaporation, samples were dissolved in 20 μL 0.1 % formic acid for LC-MS/MS analysis[56]. Peptides were eluted and separated by liquid chromatography on a Dionex system coupled via ESI to a MS consisting of a quadrupole and an orbitrap mass analyser (Orbitrap QExcactive, Thermo Scientific, Bremen, Germany). A UPLC column with a linear gradient from 2 to 30% acetonitrile/0.1% formic acid in 120 min was used. Mass spectra were measured in the positive ion mode. LC–MS/MS analysis was done on MS level over a m/z range from 400–1500. The LC-MS/MS data were processed and label free quantification (LFQ) was performed based on peak area (Proteome Discoverer 2.4.1.15; Thermo Scientific, Bremen, Germany). Identification of the proteins from the MS/MS spectra were performed with the search engine Sequest HT using the homo sapiens SwissProt database (www.uniprot.org) and a contaminant database. For each target protein, LFQ values of control samples were subtracted from those of the experimental samples (LRR domain or full length Densin-180 protein). Ranking of top targets, based on this calculation, is shown in Supplementary Table 4.

Yeast two hybrid screening was performed against a human brain cDNA library in pACT2 using protocols obtained from the manufacturer of the library (Clontech), with the reconstitution of the *His3* gene as well as β-Galactosidase staining as selectable markers in the CG1945 strain[57].

### *C. elegans* culturing, strains and CRISPR/Cas9 editing
*C. elegans* strains were cultured on nematode growth medium agar plates, with *E. coli* strain OP50 as the food source, at 20 °C, unless indicated. All strains used in this study are found in Supplementary Table 1; VC2010 was the genetic background strain employed for generating the variant lines and the deletion allele. To model the *LRRC7* missense variants L221M and L296P, each variant was knocked into the corresponding residue of the endogenous *let-413* gene (L173M and L248P, respectively) using the *dpy-10* co-conversion method of CRISPR/Cas9 editing[58,59] as previously described[60]. The editing approach uses synonymous sequence changes to generate a restriction enzyme cleavage site for variant allele genotyping and to block Cas9 re-cleavage following homology directed repair. Therefore, as controls we generated lines that contained only the synonymous changes, but not the missense variants (indicated as L173L and L248L). A deletion allele, *udn71*, was also recovered in the screen to obtain L173M edits. *Udn71* hereafter called *let-413*[Del] contains a 446-base deletion plus a 17-nucleotide insertion, starting at nucleotide 557 of the spliced F26D11a.1 transcript. The encoded protein from *udn71* contains the first 143 amino acids of *let-413* followed by 13 novel amino acids (LIEIVCIEIVCTI) and a premature stop, resulting in a product that is predicted to undergo nonsense mediated mRNA decay and considered as a putative null allele. Both *let-413*[L248P] and *let-413*[Del] are homozygous embryonic lethal and are propagated as heterozygote in trans to the inversion balancer *tmC3*[61]. All strains were backcrossed to VC2010 at least twice and the complete gene was Sanger sequenced to verify the gene structure. The guide RNAs and repair templates used are in Supplementary Table 2.

### *C. elegans* phenotypic analysis and imaging
Nomarski light microscopy, and fluorescent microscopy, were used to determine the embryonic arrest phenotype. The genotype of the examined embryos was inferred from the presence or absence of arrest phenotypes based on the following Mendelian genotype-phenotype behaviour. Of 50 self-progeny embryos from a *let-413*[L248P]/*tmC3* mother, 12 (24%) failed to hatch while the remaining 38 developed into normal adults; for seven of the unhatched embryos that were individually PCR genotyped, all were homozygous for the L248P variant. Of 50 self-progeny embryos from a *let-413*[Del]/*tmC3* mother, 14 (28%) failed to hatch while the remaining 36 developed into normal adults; for four of the unhatched embryos that were individually PCR genotyped, all were homozygous for the *udn71* deletion. AJM-1::GFP, which encodes an adherens junction component, was used to assess apical-basal epidermal cell polarity in *let-413* variant containing embryos[2,31]. Confocal images of embryos were taken with a Lecia SP8X tandem scanning confocal microscope with a white light laser using a 63x PlanApo objective, 2x zoom, and a pinhole size of 1.00 (Leica Microsystems).

### Cell culture and transient transfection
293T human embryonic kidney cells from ATCC were maintained in Dulbecco's modified Eagle's medium containing 10% foetal bovine serum and 1% penicillin/streptomycin. Cells were transiently transfected with Turbofect Transfection Reagent (Thermo Scientific) according to the manufacturer's instructions. Cells were regularly monitored for absence of mycoplasma.

### Immunoprecipitation experiments from transfected cells
Transfected cells were lysed in immunoprecipitation (IP) buffer (50 mM Tris-HCl, pH 8, 120 mM NaCl, 0.5% NP40, 1 mM EDTA), followed by centrifugation at $20,000 \times g$ for 15 min at 4 °C. GFP-tagged proteins from supernatants were immunoprecipitated from supernatants using 20 μl of magnetic GFP-trap beads (Chromotek, Munich, Germany). After washing, precipitates and input samples were processed for Western blotting.

### SDS PAGE and western blot
Proteins were separated on SDS-PAGE under denaturing conditions and blotted on nitrocellulose membrane using a MINI PROTEAN II system (Bio-Rad). After blocking membranes with 5% milk powder/TBS-T, primary antibodies in milk/TBS-T were incubated overnight at 4 °C. After washing in TBS-T, HRP-linked secondary antibodies were incubated at room temperature for 1 h. Membranes were washed and then processed for chemiluminescence; imaging was done using a ChemiDoc™ MP Imaging System (Bio-Rad). Images were processed and further analysed using Image Lab Software (Bio-Rad).

### Animals
For primary neuronal cultures, hippocampal tissue was isolated from *Rattus norvegicus* embryos. Rats (Wistar Unilever outbred rat; strain name HsdCpb:WU; were obtained from Envigo (Horst, The Netherlands). Animals were kept at Forschungstierhaltung of University Medical Center Hamburg-Eppendorf. Timed pregnant rats (4–5 months old; 250–300 g); housed in individual cages; access to food and water *ad libitum*) were sacrificed on day E18 of pregnancy using $CO_2$ anaesthesia, followed by decapitation. Neurons were prepared from all embryos (14–6 embryos, male and females); neurons from the whole preparation were pooled. 15 preparations/pregnant rats with a total of 225 embryos were used for the current study[62]. All animal experiments were approved by and conducted in accordance with the guidelines of the Animal Welfare Committee of the University Medical Center Hamburg-Eppendorf (Hamburg, Germany) under permission number Org766.

## Neuron culture and transfection

The hippocampal tissue was dissected, and hippocampal neurons were extracted by enzymatic digestion with trypsin, followed by mechanical dissociation. Dissociated neurons were plated on glass coverslips and maintained in Neurobasal medium supplemented with 2% B27, 1% Glutamax and 1% Penicillin/Streptomycin. Neurons were transfected on DIV7 using the calcium phosphate method.

## Immunocytochemistry

Neurons (DIV14) were fixed with 4% paraformaldehyde in PBS for 15 min and permeabilized with 0.1% Triton X-100 in PBS for 5 min. After 1 h blocking with 10% horse serum in PBS, cells were incubated with corresponding antibodies overnight followed by washing and then 1 h of incubation with Alexa Fluor coupled secondary antibodies. The coverslips were mounted onto glass microscopic slides using ProLong™ Diamond Antifade mounting medium.

## Microscopy

Confocal images were acquired with Leica Sp5 and Olympus FV3000 confocal microscopes. Quantitative analysis of images was performed using ImageJ. Primary dendrites were counted at a ring within 10 μm distance from the soma. The counting of clusters along dendritic branches was performed using the Multi-Point tool of ImageJ. For counting Densin-180 clusters, dendritic segments beginning at a minimum of 20 μm radial distance from the soma were analysed. Only clusters along the dendrite (distance to dendrite <3 μm) were counted. For the quantification of postsynaptic Densin-180 clusters, only Densin-180 clusters distinctly colocalizing with the postsynaptic marker Shank3 were counted. To further analyse the colocalization of Densin-180 variants with Shank3, the Pearson correlation coefficient (PCC) of green (Densin-180) and far-red (Shank3) channels of various dendritic segments was measured using the JACoP plugin of the ImageJ software. For each analysis, at least 15 neurons from three different batches were analysed, with the means of three dendritic segments from one neuron constituting one data point.

## Sholl analysis

Overview images of neurons positive for GFP-tagged Densin-180 were subjected to quantification of neuronal arborisation. Semi-automated Sholl Analysis was used, according to references[63,64]. Briefly, 8-bit images of neurons were semi-manually traced using ImageJ (NIH, Bethesda, MD). A skeletonised binary image consisting of a single neuron was then used for analysis by the open-source Sholl analysis programme[65,66] for ImageJ/Fiji. Default parameters used for the analysis were: sampling interval between radii of consecutive circles kept fixed to 10 μm and the ROI was selected in the neuronal soma. The arbours were reported as the intersections over a distance of 200 μm. For each condition, five neurons in a similar environment were used from different biological repeats.

## Reporting summary

Further information on research design is available in the Nature Portfolio Reporting Summary linked to this article.

## Data availability

The mass spectrometry proteomics data have been deposited to the ProteomeXchange Consortium (http://www.proteomexchange.org/) via the PRIDE partner repository (https://www.ebi.ac.uk/pride/) with the dataset identifier PXD044289. Data are available at http://www.ebi.ac.uk/pride/archive/projects/PXD044289. Genome sequencing data from the 100KGP have been deposited in the National Genomic Research Library at https://www.genomicseducation.hee.nhs.uk/supporting-the-nhs-genomic-medicine-service/national-genomic-research-library-information-for-clinicians/. All researchers are vetted by an oversight committee before being granted permission to access the data, which is held in a form that cannot be copied or removed without the permission of Genomics England, and may only be used for purposes that are in line with their acceptable use policy. Access to datasets referenced in the Methods section is provided via https://www.genomicsengland.co.uk/research/research-environment. Data from the Undiagnosed Disease Network (UDN) are available in the dbGaP database under accession code phs001232.v5.p2 (https://www.ncbi.nlm.nih.gov/projects/gap/cgi-bin/study.cgi?study_id=phs001232.v5.p2). The data are available by dbGaP Authorized Access overseen by the National Human Genome Research Institute's data access committee. Investigators must apply to access the data. Investigators and their institutions will be required to agree to the Data Use Certification Agreement prior to approval for data access. In addition, raw sequencing data for additional patients cannot be deposited in public databases because the contributing Institutions do not have consent for this. Therefore, access should be requested from the corresponding author, who will contact the respective Institutions and respond within a month. All other data supporting the findings described in this manuscript are available in the article and the Supplementary Information, Supplementary Data and Source Data file. Source data are provided with this paper.

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

## Acknowledgements
We thank UKE microscopic imaging facilty (umif) for providing confocal microscopes. Purchase of the Olympus microscope was supported by Deutsche Forschungsgemeinschaft (DFG grant: INST 152/933-1). This research was made possible through access to data in the National Genomic Research Library, which is managed by Genomics England Limited (a wholly owned company of the Department of Health and Social Care). The National Genomic Research Library holds data provided by patients and collected by the NHS as part of their care and data collected as part of their participation in research. The National Genomic Research Library is funded by the National Institute for Health Research and NHS England. The Wellcome Trust, Cancer Research UK and the Medical Research Council have also funded research infrastructure. Research reported in this manuscript was also supported by Deutsche Forschungsgemeinschaft (Kr1321/9-1, to H.J.K.; SCHL2276/2-1 and 450149205-TRR333/1 to Ch.S.; 458949627; ZE 1213/2-1 to M.Z.); by Jung-Stiftung für Wissenschaft und Forschung (to A.L.B.); and the NIH Common Fund, through the Office of Strategic Coordination/Office of the NIH Director under Award Number U01HG007672 (PI: Shashi V, Duke University) and U54NS108251 (PIs: Schedl T & Solnica-Krezel L, Washington University). The content is solely the responsibility of the authors and does not necessarily represent the official views of the National Institutes of Health. M.Z. receives research support from the EJP RD (EJP RD Joint Transnational Call 2022); from the German Federal Ministry of Education and Research (BMBF, Bonn, Germany), awarded to the project PreDYT (PREdictive biomarkers in DYsTonia, 01GM2302); from the Federal Ministry of Education and Research (BMBF) and the Free State of Bavaria under the Excellence Strategy of the Federal Government and the Länder; from the Technical University of Munich—Institute for Advanced Study; and through a 'Schlüsselprojekt' grant from the Else Kröner-Fresenius-Stiftung (2022_EKSE.185). One of the authors of this publication is a member of the European Reference Network on Rare Congenital Malformations and Rare Intellectual Disability ERNE-ITHACA. (EU Framework Partnership Agreement ID: 3HP-HP-FPA ERNE-01-2016/739516).

## Author contributions
J.W., D.W., S.D., K.P., A.L.B., D.T., H.-H.H., F.H.N. and H-J.K. performed experiments in cells and neurons. W.Y., A.L. and O.I. performed *C.elegans* experiments. L.F., Ar.S., An.V., Th.S., C.C., K.B., Kr.S., A.B., Ke.S., R.S., E.D., E.C., F.V., K.P., S.N., J.G.-A., A.N.G, B.M., Al.S., M.S., N.M., Ca.S., M.C.-D., J.H., J.V.-G., J.-M.d.S.A., M.T., E.M.B., M.W.-H., A.-C.T., Al.V., J.L., Ch.S. and X.L. were involved in patient care, analysed genetic data, and gathered detailed clinical information for the study. G.A.S., S.C.P. and Ti.S. directed and analysed *C.elegans* experiments. M.Z. and T.B. analysed trio-exome sequencing data. S.H. performed mass spectroscopy analysis. D.G., K.F., A.M. and E.T. analysed data from the 100 k genomes project. Ti.S. and V.S. coordinated work by the Undiagnosed Disease Network and acquired funding. H-J.K. acquired funding, coordinated the study and drafted the initial manuscript.

## Funding

## Competing interests
The authors declare no competing interests.

## Additional information

Jana Willim[1], Daniel Woike[1], Daniel Greene[2], Sarada Das[1], Kevin Pfeifer[1], Weimin Yuan[3], Anika Lindsey[3], Omar Itani[3], Amber L. Böhme[1], Debora Tibbe[1], Hans-Hinrich Hönck[1], Fatemeh Hassani Nia[1], Undiagnosed Diseases Network*, Michael Zech[4,5,6], Theresa Brunet[4,5], Laurence Faivre[7,8], Arthur Sorlin[8,9], Antonio Vitobello[8,9], Thomas Smol[10], Cindy Colson[10], Kristin Baranano[11], Krista Schatz[12], Allan Bayat[13,14,15], Kelly Schoch[16], Rebecca Spillmann[16],

Erica E. Davis [17,18,19], Erin Conboy[20], Francesco Vetrini[20], Konrad Platzer [21], Sonja Neuser [21], Janina Gburek-Augustat [22], Alexandra Noel Grace[23], Bailey Mitchell[24], Alexander Stegmann [25], Margje Sinnema[25], Naomi Meeks[26], Carol Saunders [27,28,29], Maxime Cadieux-Dion[27], Juliane Hoyer[30], Julien Van-Gils [31], Jean-Madeleine de Sainte-Agathe[31], Michelle L. Thompson[32], E. Martina Bebin[33], Monika Weisz-Hubshman[23,34], Anne-Claude Tabet[35], Alain Verloes [35], Jonathan Levy [35], Xenia Latypova[35], Sönke Harder [36], Gary A. Silverman [3], Stephen C. Pak [3], Tim Schedl[37], Kathleen Freson[38], Andrew Mumford[39], Ernest Turro [2], Christian Schlein [1], Vandana Shashi[16] & Hans-Jürgen Kreienkamp [1] ✉

[1]Institute of Human Genetics, University Medical Center Hamburg-Eppendorf, Hamburg, Germany. [2]Icahn School of Medicine at Mount Sinai, New York, NY, USA. [3]Department of Pediatrics, Washington University in St Louis School of Medicine, St Louis, MO, USA. [4]Institute of Human Genetics, School of Medicine, Technical University of Munich, Munich, Germany. [5]Institute of Neurogenomics, Helmholtz Zentrum München, Munich, Germany. [6]Institute for Advanced Study, Technical University of Munich, Garching, Germany. [7]Centre de Génétique et Centre de Référence Anomalies du Développement et Syndromes Malformatifs, FHU TRANSLAD, CHU Dijon-Bourgogne, Dijon, France. [8]INSERM—Université de Bourgogne—UMR1231 GAD, Dijon, France. [9]Laboratoire de Génomique médicale, Centre NEOMICS, CHU Dijon Bourgogne, Dijon, France. [10]Univ. Lille, CHU Lille, ULR7364 – RADEME, Lille, France. [11]Department of Neurology, Johns Hopkins University School of Medicine, Baltimore, MD, USA. [12]Department of Genetic Medicine, Johns Hopkins University School of Medicine, Baltimore, MD, USA. [13]Department of Epilepsy Genetics and Personalized Medicine, Danish Epilepsy Center, Dianalund, Denmark. [14]Department for Regional Health Research, University of Southern Denmark, Odense, Denmark. [15]Department of Drug Design and Pharmacology, University of Copenhagen, Copenhagen, Denmark. [16]Division of Medical Genetics, Department of Pediatrics, Duke University School of Medicine, Durham, NC, USA. [17]Center for Human Disease Modeling, Duke University Medical Center, Durham, NC, USA. [18]Stanley Manne Children's Research Institute, Ann & Robert H. Lurie Children's Hospital of Chicago, Chicago, IL, USA. [19]Departments of Pediatrics and Cell and Developmental Biology, Feinberg School of Medicine, Northwestern University, Chicago, IL, USA. [20]Indiana University School of Medicine, Indianapolis, IN, USA. [21]Institute of Human Genetics, University of Leipzig Medical Center, Leipzig, Germany. [22]Division of Neuropaediatrics, Hospital for Children and Adolescents, University of Leipzig Medical Center, Leipzig, Germany. [23]Molecular and Human Genetics Department, Baylor College of Medicine, Houston, TX, USA. [24]Baylor College of Medicine in San Antonio, San Antonio, TX, USA. [25]Department of Clinical Genetics, Maastricht University Medical Center, Maastricht, The Netherlands. [26]Children's Hospital Colorado, Division of Clinical Genetics & Metabolism, Aurora, CO, USA. [27]Department of Pathology and Laboratory Medicine, Children's Mercy Hospital, Kansas City, MO, USA. [28]School of Medicine, University of Missouri Kansas City, Kansas City, MO, USA. [29]Genomic Medicine Center, Children's Mercy Research Institute, Kansas City, MO, USA. [30]Institute of Human Genetics, Friedrich-Alexander-Universität Erlangen-Nürnberg, Erlangen, Germany. [31]Genetics Lab, Centre Hospitalier Universitaire (CHU) de Bordeaux, Bordeaux, France. [32]HudsonAlpha Institute for Biotechnology, Huntsville, AL, USA. [33]University of Alabama at Birmingham, Birmingham, AL, USA. [34]Texas Children's Hospital, Houston, Tx, USA. [35]Department of Genetics, APHP-Robert Debré University Hospital, Paris, France. [36]Mass spectrometry and Proteome Analytics, Institute for Clinical Chemistry and Laboratory Medicine, University Medical Center Hamburg-Eppendorf, Hamburg, Germany. [37]Department of Genetics, Washington University in St Louis School of Medicine, St Louis, MO, USA. [38]Department of Cardiovascular Sciences, Center for Molecular and Vascular Biology, KU Leuven, Leuven, Belgium. [39]School of Cellular and Molecular Medicine, University of Bristol, Bristol, UK. **A list of authors and their affiliations appears at the end of the paper. ✉ e-mail: Kreienkamp@uke.de

## Undiagnosed Diseases Network

Kelly Schoch[16], Rebecca Spillmann[16], Monika Weisz-Hubshman[23], Stephen C. Pak [3], Tim Schedl[37] & Vandana Shashi[16]

