## [Peer Review File · Nature Communications]

Variants in LRRC7 lead to intellectual disability, autism, aggression and abnormal eating behaviorsREVIEWER COMMENTS

Reviewer #1 (Remarks to the Author):

This manuscript describes the identification of several mutated variants of the LRRC7 gene, which encodes densin scaffolding proteins that are well known to be highly expressed at excitatory synapses in neurons. Notably the probands bearing these mutations share an overlapping set of behavioral abnormalities, including intellectual disabilities. In an effort to demonstrate that these mutations functionally impact densin and may contribute to the behavioral changes, the authors perform numerous protein-protein interaction assays to suggest that the mutations may differentially impact the multi-protein complexes that densin may be able to assemble in neurons. Significantly, they use BioID and yeast 2-hybrid approaches to identify for the first time other proteins that appear to bind to the leucine-rich repeat domain of densin. Overall, these studies are very interesting and the findings have the potential to be quite impactful. However, the current manuscript has a number of significant weaknesses (see below) that together mean that it falls some way short of conclusively supporting the interpretation that "rare variants in LRRC7, coding for Densin-180, lead to a neurodevelopmental disorder in humans", as stated in the first line of the Discussion.

1. The protein-protein interaction assays rely exclusively on over-expressing tagged proteins in heterologous cells. There is scant evidence that the newly identified putative interaction partners (e.g., PP1) actually associate with densin in neurons when the proteins are expressed at endogenous levels. Related to this, the BioID screening for interacting partners uses HEK293 cells, but this could easily miss significant neuronal interaction partners that are not expressed in these cells.

2. The methods used for the mass spectrometry analyses are not coherently described in sufficient detail in the manuscript, the supplement or in the ProteomeXchange database record to evaluate the quality of the analyses. How long were the cells labeled with biotin and at what temperature? It reads to me like the authors prepared an entire gel lane containing all the proteins for their shotgun analysis, but this should be more explicitly stated. What do the authors mean by "evaluating the signal strength in Densin samples vs control samples" (Supplemental Results) – spectral counts? I think the main manuscript should provide some indication of this quantitation for non-specialist readers so that the veracity of the proteins listed in Supplemental Table 4 can be evaluated without having to dig through the detailed data sets that are deposited in the database.

3. There is no effort to define the biological impact of disrupting any of these protein protein interactions in neurons (or animals). The authors could examine whether expressing the mutated densins has an impact beyond having generally little effect on the number of primary dendrites. For example, it should be fairly straightforward to compare the impact of expressing the densin variants on overall neuronal morphology (e.g., dendritic arborization) and the total number of synapses, marked by a bone fide synaptic marker.

4. Representative images examining the localization of densin variants and Shank3 indicate that the neurons expressing densin (WT or variants) are in very different environments – sometimes the only visible cell in the field of view (e.g., Fig. 4C top panel or Fig. 5A second row), but in other images embedded within a cluster of nearby neurons (e.g., Fig. 4C bottom and Fig. 5A third row appear to have at least 10 nearby neurons). This not only affects the amount of background staining of endogenous proteins in non-transfected neurons (e.g., Shank3) that could impact the colocalization analysis, but might also affect the amount of intrinsic activity in the cells, potentially impacting their development and morphology. Related to this, the methodological description reporting of data from the dendritic cluster analysis (densin and co-localization with Shank3) lack sufficient detail to evaluate. My impression is that each of the data points superimposed on bar graphs in Figs. 4-6 represents the number of puncta on a single dendrite, and that the authors analyzed perhaps 10-20 dendrites in most of these datasets. How many neurons were analyzed to accumulate these data? How many batches of neurons were transfected and did analysis of each batch reveal similar differences?

5. The authors over-interpret Supplemental Fig. 7 as supporting the formation of what they refer to a "Triple Complex" containing densin, PP1 and CaMKII. far more detailed studies are required to better substantiate such a claim since sub-stoichiometric amounts of PP1 and CaMKII could easily be interacting with different densin molecules.

6. In the discussion, the authors conclude that "most of the binding to α CaMKII is lost upon deletion of the last 350 amino acids (from densin), but some specific binding remains, likely at the central site." However, it does not appear that the activation of CaMKII was controlled in these assays, and it is important to note that the central site only binds to the activated kinase.

7 The authors may wish to consider how the truncated densin proteins that may be expressed by some of the mutated DNAs relate to the numerous mRNA splice variants of the rat densin characterized previously (Jiao et al., 2008. PMID: PMC2814316).

8. I do not have the expertise to properly evaluate the patient and genomic data

Reviewer #2 (Remarks to the Author):

The manuscript by Willim and colleagues presents LRRC7 as a new disease-gene. The authors provide sufficient clinical and molecular evidence to demonstrate that variants in LRRC7 lead to the onset of a neurodevelopmental disorder mainly characterized by speech delay, intellectual disability, and aggressive behavior. The manuscript is well written and merits publication. Despite this interesting content, the manuscript would benefit from revisions. The following specific points need to be addressed:

- In the introductions, lines 92-93, the authors mentioned that a single patient with a loss-of-function variant in LRRC7 was observed in an exome sequencing study of NDD patients (Popp 2017). However, there are two additional frameshift variants reported in HGMD that should be also mentioned: p.Lys120Valfs*2 (Yuen 2015, PMID: 25621899) and p.Val1032Serfs*5 (Fu 2022 and Zhou 2022, PMID: 35982160, 35982159)

- In the introduction (line 94), the authors state that they collected a group of 30 patients. In the abstract (line 63), 31 patients are mentioned. The authors should address this discrepancy throughout the manuscript

- In the results paragraph "identification of LRRC7 variants in patients with a neurodevelopmental disorders", the cohort identified from the 100 KGP and Genematcher are discussed separately. I understand that it makes sense to explain how the cohorts were assembled, but it would be less confusing if the patient mutations and clinical features would be discussed all together instead of separately. For example: "in total, we identified 31 patients with 24 different variants with the following clinical features..."

- The authors speculate that the father of case 11 is likely carrier of the LRRC7 variant because both younger half siblings on the father's side are mentally retarded (I would avoid using this term in the supplementary patients' descriptions) and because the father displayed aggressive behavior and ID. However, in the absence of genetic testing in the father or in the two half siblings, it cannot be excluded that the reasons for the ID of the siblings and father might be different. If the authors want to provide an example of carrier parent with some degree of neurological manifestations, they should discuss a parent for whom genetic testing has been performed

- At the moment, the clinical features of the patients are scattered across Figure 1, Supplementary Figure 1, the supplementary table and the supplementary text. I believe the manuscript would benefit if there would be a table summarizing the main clinical features of the LRRC7 patients and the total frequency of each feature (expressed in percentage) in the main text.

- Have the authors attempted to compare the clinical features of the patients with truncating variants with those of the patients with missense variants? Is there any difference that would help to phenotypically distinguish the two classes of patients?

- The functional analysis of the missense variants in *C. elegans* (line 298) was only performed on mutations p.L211M and p.L296P. Is there any reason why specifically these two mutations were chosen and not the other four mutations in the LRR repeats (L65S, N94S, C106Y, and A139P)? Why was the analysis limited to these two mutations and not extended to the other mutations?

- I believe that the statement "Nonsense variants disrupt C-terminal interactions of Densin-180" (which is also the title of one of the results paragraph) is incorrect. As the authors themselves state, nonsense and frameshift variants are likely subject to nonsense mediated decay, which would lead to degradation of the mutant transcript. Hence, the truncated proteins are likely not produced in the cells. This applies also to the two tested mutations p.R1018* and p.Y1210* (both fall within exon 21 of 27 and are therefore likely associated with NMD). For this reason, the in vitro generation of the two resulting truncating proteins does not inform on the real pathogenic mechanism associated with truncating variants. This experiment exclusively shows that the PDZ domain of Densin-180 is important for the interaction with α -CaMKII and δ -catenin, which was previously already shown. Hence, the real pathogenic mechanism associated with truncating variants (frameshift and nonsense) is still unknown and can be investigated only in the presence of patients' cells. Thus, the statement "Nonsense variants disrupt C-terminal interactions of Densin-180" is incorrect: the pathogenic mechanism of these variants is likely haploinsufficiency rather than due to impaired interaction

- Why was the spine-shaft ratio of GFP-fluorescence intensities not measured for the mutations N94S, C106Y, and A193P (Fig 6)? I believe all the mutations should be assessed using the same parameters

- In supplementary Figure S6, two different clones of MACF1 are used. What is the difference between the two clones? Is the cloned fragment exactly of the same size in both clones? Were the plasmid sequences verified by Sanger? The authors should also specify the size of the MACF1 fragments used (from amino acid XX to amino acid XX of MACF1)

- Several analyses have been performed to assess the pathogenicity of the missense variants within the LRR domain (functional analysis in *C. elegans*, dendrite morphology, post synaptic targeting, interaction with α -CaMKII, δ -catenin, Shank3, MACF1, Erbin, PP1 α). I think the manuscript would benefit if there would be a table in the main text summarizing the analyses done and specifying which analysis/parameter is altered for each specific LRR-missense variant. In this way, it would become immediately apparent which specific Densin-180 function is altered in the presence of each mutation

Minor comments:

- In Figure 6, the legend letter D is not visible in the figure

- In the legend of Figure 4, specify threshold for significance (***, significantly different from wt: which is the threshold?)

- In Supplementary Figure 5, the mutation L221M is wrongly indicated as L221P in graphs B, D, and F. In addition, in the Western blot of part A, the band corresponding to L221M is not detected, neither in the input, nor in the immunoprecipitated fraction. Do the authors have an additional western blot picture where also L221M signal is visible? How many times was the analysis performed? (n=3?)

Reviewer #3 (Remarks to the Author):

This is a well written study on association of the LRRC7 deficiency with human disease where intellectual disability, neurodevelopmental delay and abnormal behavior are some of the common features. In total, 31 individuals were identified with heterozygous (predominantly loss of function) variants in LRRC7. Further molecular studies showed that some of the (patient) missense variants in the PDZ domain interfere with targeting densin to neurons, while variants in the LRR domain interfere with binding of the LRRC7 interaction partners that were identified in this manuscript. Modeling using *C. elegans* model organism showed that one of the missense variants (L248P), identified in patients, is pathogenic (i.e. leads to embryonic lethality). Overall, this is an extensive cohort of patients where various molecular assays were conducted to confirm the role of LRRC7 in neurodevelopment. The work is important and of good quality.

Comments to the author:

Major comments

- In terms of the patient cohort, it is difficult to follow which patients make the cohort of 31 described here; how many come from which study and how many belong to the same family. There are 21 case reports in the supplemental and 21 in the supplemental Excel table; it would be helpful to explain it a bit clearer how this number relates to 31.
- Detailed description and main text discussion is necessary for the families where variants are inherited from "unaffected" parents. Do these variants tend to be less severe, missense? Do phenotypes in these families tend to be milder? Discussion on potential reduced penetrance observed with other disorders related to LRRC7 would strengthen the manuscript.
- Lines 272-274 "This is exemplified ..." It is not clear what is the basis of the conclusion that father who is unavailable for testing is likely a carrier. Please explain and provide evidence.
- Considering high phenotypic variability between patients, it is necessary to include information on other variants in the exomes/genomes. Were exome/genome data checked for other contributing variants (i.e. dual hits/diagnoses), especially in more severe cases?
- In terms of *C. elegans* work, it is unclear why the authors used the *udn71* allele rather than generating a targeted knockout (complete ORF deletion). It seems that the *udn71* allele was generated as a byproduct of CRISPR, unintentionally. It is a bit worrisome how the *udn71* was obtained as this genome may contain additional unintentional variants. It would be preferential to obtain a deletion allele where entire ORF is deleted.
- Also, why focus only on the p.L211M and p.L296P variants for modelling in *C. elegans*? Were these the only variants conserved well enough to allow for the knock-in CRISPR?
- Were any attempts made to further characterize the L173M (L211M) variant? Is it temperature sensitive? Any subtle defects using *AJM::GFP*? How about seam cell phenotypes?
- In addition to *C. elegans* work, the L211M when tested using other methods does not seem to show any obvious effects. How did the authors then conclude that L211M is mild rather than benign variant? The same question also applies to A193P (line 400).
- Given that different molecular tests were done on different LRRC7 variants, it is necessary to include a summary table to summarize for how many of the variants were validation tests attempted, for how many not, for how many only one test, for how many the experimental tests indicate potential effect and for how many benign status. It would be also useful to use ACMG guidelines to classify the variants. Will the variants be submitted to ClinVar?
- Also, I could not find anywhere in the manuscript population frequency of these variants (e.g. in gnomAD and TopMed databases), please include.
- In terms of phenotype, it seems that the single overlapping feature between the patients is abnormal nervous system physiology/abnormality of the nervous system, but it is not explained what that is and what kind of testing was used to determine this phenotype.
- Hyperphagia does not appear to be included in the graphic presentation of phenotypes.

Minor comments

- Genematcher is misspelled throughout, please revise to GeneMatcher.

Reviewer #4 (Remarks to the Author):

This study describes the discovery and characterization of rare variants in the LRRC7 gene, which codes for the LAP protein Densin-180, and their association with a neurodevelopmental disorder in humans. The study combines data from the 100k genomes project with patient data from genetic laboratories. In total, 31 individuals with LRRC7 variants are identified, showing a core phenotype of intellectual disability, delayed language development, ADHD, autism, and seizures in some cases. Aggressive or impulsive behavior is also common among these patients.

The functional impact of loss-of-function and missense variants affecting the LRRC7 gene are explored using biochemical, genetic (*C. elegans*), cellular, and imaging approaches. Patient variants in LRRC7 variants disrupt interactions with known partners, such as δ -catenin, and newly identified interacting proteins (Erbin and MACF1). PP1 α is also identified as an important interaction partner of Densin-180, indicating a potential role in regulating synaptic signaling. The authors discuss the importance of the Densin-180/PP1 complex in synaptic plasticity and the potential impact of LRRC7 variants on this process.

Major comments:

- Regarding the functional validation studies in *C. elegans*:
 - Patient variants are heterozygous. The authors indicate that let-413 mutants are recessive. Are they fully recessive or are there also hints of haploinsufficiency in *C. elegans*?
 - Along these lines, the authors conclude that L221M is a mild mutation based on very limited impact in different assays and no obvious impact in *C. elegans*. Maybe some subtle perturbations of the AJM-1::GFP pattern could provide confirming evidence about the pathogenicity of this variant.
 - In fact, in the absence of stronger evidence about its functional impact, wouldn't it be prudent to reclassify the variant as VUS, pending additional functional data? This conclusion may have important clinical implications for the patient, potentially prompting further genetic analysis.
- Statistical analysis of Western/IP quantifications.

Student's t-tests are applied in Figures 4B,8B,8D,8F,S2,etc., to compare the quantified Western/IP data. Applying a Student's t-test to this data is not possible, as there are only three data points. Furthermore, post-hoc multiple-comparisons tests also need to be applied to correct these analyses. Please correct these issues throughout the manuscript.
- Regarding Figure 4A,B:
 - It is unclear whether the three samples presented in Figure 4A are independent biological replicates. Please provide this type of information throughout the manuscript, whenever applicable.
- Regarding Figure 4C-F:
 - The distribution of Shank3 appears very different in the L1567Q mutant. Is there data ensuring that synapse formation is not altered in general upon L1567Q expression in this cellular context? Indeed, an alternative hypothesis could be that the over-expression of the L1567Q mutant alters the synaptic organisation in general or the distribution of Shank3 specifically.
- Regarding Figure 4C-F and Figure 6:
 - Methodological information is very limited regarding these experiments. For instance, how are Densin-180 clusters counted? From how many independent experiments is the data derived (independent transfections, cells, dendrites, etc.). Reading the Methods section, it seems they are counted "manually". What kind of image treatment was applied (e.g., thresholding, segmentation) to determine whether a group of pixels indeed constitutes a "synaptic cluster"?

Regarding the enrichment of Densin-180 at Shank-3-labeled sites, Pearson correlation analyses are generally used to quantify synaptic co-localisation patterns. See for example Verstraelen P, et al., iScience. 2020). Please implement this type of analysis to determine enrichment (or not) of Densin-180 at Shank3 labeled clusters for wild-type and patient variants.

Minor comment

- Fig.2B : The letters in the "consensus" line are not aligned with the sequence of the different LRR repeats.

Reviewer #5 (Remarks to the Author):

The authors describe 31 individuals with variants in the LRRC7 gene and significant neurodevelopmental anomalies. This is a major paper defining a new human disease going well beyond describing the phenotype and genotype, to also defining mutation mechanisms and enlarging our knowledge of LRRC7 functions.

The paper is well written, concise, and clear. I do not think it should be shortened.

I have no problems with the descriptions, experiments, discussion, or data; below are some minor suggestions to consider.

Introduction

Good and interesting.

Abstract says 31 patients, Intro says 30? This is a confusing/annoying. Maybe better later stated as, "In total we describe 31 individuals with variants in LRRC7.", as one is from the literature.

Materials and methods

I cannot comment on the C. Elegans work methods or results.

The HEK 293T cell line is called both 293T cells and 293T HEK – please harmonise, e.g. line 547 in Figure legends.

Results

"In many cases this is associated with delayed speech and language development as well 253 as delayed fine motor development. Several patients were affected by autism spectrum disorder (ASD) 254 as well as ADHD. Furthermore, selective mutism and impulsive and aggressive or auto-aggressive 255 behaviour was reported..." – could numbers with each feature be shown in brackets? Later such numbers are given in the paragraph, e.g. for brain malformations.

"Four patients with truncating 256 variants in the central part of the mRNA/protein showed hyperphagia due to reduced satiety, leading 257 to class II obesity in three of them." Were these the only patients with such variants - if so, can that be stated please.

"Again, loss-of-function variants (nonsense, splice site and frameshift variants, and a 99 kB deletion 261 encompassing the last exon), were observed (13 patients)." Again is not needed.

It is cumbersome that the phenotypes and genotypes of each group is described separately, and not collectively. Whilst I understand why the paper was written this way, combining the results would be much easier for the clinician or scientist who will read the paper.

"Seven pedigrees were identified with inherited cases, indicating that LRRC7 variants display incomplete penetrance."

May be better to be,

"Seven pedigrees were identified with inherited cases, indicating that LRRC7 variants may display incomplete penetrance."

Paragraph from lines 280 to 297; the descriptions of the amino acid changes are valid (and interesting to this Reviewer), but inclusion of a known score of pathogenicity, say from AphaSense, may enhance readability.

Line 400 "C. elegans embryo (Fig. 3). Thus, these two appear to be rather mild mutations." Did this coincide with a milder phenotype?

Discussion

Line 418, "...of the 100k genomes project... you have previously defined this as 100KGP, so why not use the abbreviation?"

Line 92. "In addition, a single patient with a loss-of-function variant in LRRC7 was observed in an exome sequencing study of NDD patients (20)."

To

"In addition, a single patient with a loss-of-function variant in LRRC7 was observed in an earlier exome sequencing study of NDD patients (20)."

Line 426 "Furthermore, aggressive or impulsive behaviour is prevalent in many cases reported here." Could you comment on any sex effect of this behaviour in the 30 cases, and state in how many cases this could be judged, i.e. if very young, aggression maybe hard to assess.

Line 440, "We noted that in several cases, these variants were inherited from healthy parents, indicating incomplete penetrance of the phenotype. However, due to the complexity of the phenotype, it appears possible that specific aspects of e.g. aggression or obesity may have been overlooked in the parents."

It was unclear how many parents had been properly/completely phenotyped, and this should be stated before any comment is valid about incomplete penetrance. A more nuanced approach maybe to state that it is unclear if the condition always manifests, and omit the last sentence (as it could also apply to learning difficulties in childhood, without interviewing the respective grandparents.

Line 444 " Further insight into the functional aspects of Densin-180 required for normal brain function may be.." may be changed to "were"

Line 450 "...which were not known so far." Delete "so far"

Line 533 Figure, "AlphaFold" not "alphaFold".

Line 556 "(F). ***, significantly different from WT, t-Test." *** is not defined, and is $p < 0.001$? (Embarrassingly I looked this up, and it is either "Student's t test" or "T test", not t-Test.)

Supplementary data

The case histories are a valuable addition.

REVIEWER COMMENTS

Reviewer #1 (Remarks to the Author):

This manuscript describes the identification of several mutated variants of the LRRC7 gene, which encodes densin scaffolding proteins that are well known to be highly expressed at excitatory synapses in neurons. Notably the probands bearing these mutations share an overlapping set of behavioral abnormalities, including intellectual disabilities. In an effort to demonstrate that these mutations functionally impact densin and may contribute to the behavioral changes, the authors perform numerous protein-protein interaction assays to suggest that the mutations may differentially impact the multi-protein complexes that densin may be able to assemble in neurons. Significantly, they use BioID and yeast 2-hybrid approaches to identify for the first time other proteins that appear to bind to the leucine-rich repeat domain of densin. Overall, these studies are very interesting and the findings have the potential to be quite impactful. However, the current manuscript has a number of significant weaknesses (see below) that together mean that it falls some way short of conclusively supporting the interpretation that “rare variants in LRRC7, coding for Densin-180, lead to a neurodevelopmental disorder in humans”, as stated in the first line of the Discussion.

Response: We would like to thank all five reviewers for the positive evaluation and constructive criticism of our manuscript.

1. The protein-protein interaction assays rely exclusively on over-expressing tagged proteins in heterologous cells. There is scant evidence that the newly identified putative interaction partners (e.g., PP1) actually associate with densin in neurons when the proteins are expressed at endogenous levels. Related to this, the BioID screening for interacting partners uses HEK293 cells, but this could easily miss significant neuronal interaction partners that are not expressed in these cells.

Response: Our analysis of the native interaction partners of Densin-180 has been hampered by the absence of a reliable coimmunoprecipitation protocol. Published IP data for Densin-180, including our own work from 2005, demonstrate these difficulties as precipitating antibodies create considerable background levels in blots. This is ok for large proteins such as Shank3 (which migrate above the antibody smear), but difficult for small proteins such as PP1, which are buried below these antibodies.

In addition, work by Kennedy's group has shown that the majority of Densin-180 found in brain is insoluble in mild detergents which are typically used for IP experiments. RIPA buffer, or better 1 % desoxycholate, pH 9.0 is required for efficient solubilisation in our hands. This is not compatible with interactions which are comparatively weak and transient. Please note that the original observation of the interaction of PP1 with the LRR domain of LAP proteins (i.e. with Scribble) came from experiments using a covalent, bifunctional crosslinking reagent, which was needed to stabilize the interactions (see ref. by Troyanovski et al., 2021; ref. 47). Obviously, we can not use such an approach on interaction studies e.g. in mouse brain. So, for the time being, evidence for interaction between Densin-180 and PP1 in the brain in vivo is based on the enrichment of both proteins at the postsynapse, and the altered time course of phosphorylation of CaMKII in *Lrrc7* ko mice observed by Carlisle et al., 2011 (ref. 13).

Regarding the BioID experiment and the possibility to perform it in a neuronal system: initially we planned to do such an experiment, hoping that work in 293T cells would only be a pilot experiment.

But, reading up on this, particularly on a groundbreaking study by Uezu et al (2016; PMID 27609886; ref. 53), we made several conclusions with respect to our work:

(a) Uezu et al (2016) used BirA fusions for Gephyrin, Arhgef9 and Insyn1, for inhibitory synapses, and PSD-95 (for excitatory synapses). In each case, they pulled out a complex of more than 100 proteins, i.e. the whole inhibitory and the whole excitatory postsynaptic complex. This included the already known direct interaction partners of Gephyrin and PSD-95; but from the BioID data set it would have been impossible to deduce these direct interaction partners. Thus, our expectation was that by performing a Densin-180-BioID in neurons or brain, we would pull down the whole PSD complex. But we would not be any smarter with respect to a direct interaction partner of the LRR domain, which was needed here for the analysis of the LRR domain variants found in patients. 293T cells are of course much less complex and do not contain synaptic protein assemblies, which made it easier to identify putative direct partners of the Densin LRR domain.

(b) a surprising result of the study by Uezu et al (2016) was that Densin-180 is one of the very few proteins present in inhibitory and excitatory postsynaptic complexes. In addition, PP1 α was also found enriched at the inhibitory complex, in addition to the well-known enrichment at the excitatory PSD complex we have also observed here. We have now briefly mentioned the possible occurrence of both PP1 and Densin-180 at inhibitory synapses in the Discussion.

2. The methods used for the mass spectrometry analyses are not coherently described in sufficient detail in the manuscript, the supplement or in the ProteomeXchange database record to evaluate the quality of the analyses. How long were the cells labeled with biotin and at what temperature? It reads to me like the authors prepared an entire gel lane containing all the proteins for their shotgun analysis, but this should be more explicitly stated. What do the authors mean by “evaluating the signal strength in Densin samples vs control samples” (Supplemental Results) – spectral counts? I think the main manuscript should provide some indication of this quantitation for non-specialist readers so that the veracity of the proteins listed in Supplemental Table 4 can be evaluated without having to dig through the detailed data sets that are deposited in the database.

Response: we have now provided all details of our experimental approach in the Methods section of the main manuscript. This includes the time for biotinylation; the step where we run the samples into the gel (only into the first 5 mm; which is then cut out); and the label free quantification approach (LFQ).

3. There is no effort to define the biological impact of disrupting any of these protein protein interactions in neurons (or animals). The authors could examine whether expressing the mutated densins has an impact beyond having generally little effect on the number of primary dendrites. For example, it should be fairly straightforward to compare the impact of expressing the densin variants on overall neuronal morphology (e.g., dendritic arborization) and the total number of synapses, marked by a bone fide synaptic marker.

Response: We have now included Sholl analysis for neurons expressing all missense variants. These data indeed add new insight, as we observed for all LRR variants a slight decrease in the branching of dendrites when compared to neurons expressing the WT. This became significant for the L65S, N94S and L296P variants at some of the distance points. The PDZ domain variant did not have any effect (new supplemental Figure S4).

Furthermore, we have analysed the impact of Densin-180 protein variants on the number of synapses. We present in the revised version the number/density of Shank3 clusters on dendrites. The endogenous Shank3 protein is well established as a marker for postsynaptic sites, and in fact is used here. In a separate experiment, we have determined for one of the mutants the number of

vGlut/Shank3 coclusters, which is a more stringent indicator of true synaptic contacts. For both types of analysis, we did not observe any changes between WT and variants (new supplemental Figure S6).

4. Representative images examining the localization of densin variants and Shank3 indicate that the neurons expressing densin (WT or variants) are in very different environments – sometimes the only visible cell in the field of view (e.g., Fig. 4C top panel or Fig. 5A second row), but in other images embedded within a cluster of nearby neurons (e.g., Fig. 4C bottom and Fig. 5A third row appear to have at least 10 nearby neurons). This not only affects the amount of background staining of endogenous proteins in non-transfected neurons (e.g., Shank3) that could impact the colocalization analysis, but might also affect the amount of intrinsic activity in the cells, potentially impacting their development and morphology. Related to this, the methodological description reporting of data from the dendritic cluster analysis (densin and co-localization with Shank3) lack sufficient detail to evaluate. My impression is that each of the data points superimposed on bar graphs in Figs. 4-6 represents the number of puncta on a single dendrite, and that the authors analyzed perhaps 10-20 dendrites in most of these datasets. How many neurons were analyzed to accumulate these data? How many batches of neurons were transfected and did analysis of each batch reveal similar differences?

Response: we have now included more representative pictures for neurons in all cases, making sure that a similar number of cells is seen in each microscopic view. Regarding the number of neurons analysed: we have analysed three dendritic segments from each neuron, which then constitutes one data point. For every analysis, 15 neurons/45 dendritic segments from at least three different batches were quantitated.

5. The authors over-interpret Supplemental Fig. 7 as supporting the formation of what they refer to a “Triple Complex” containing densin, PP1 and CaMKII. far more detailed studies are required to better substantiate such a claim since sub-stoichiometric amounts of PP1 and CaMKII could easily be interacting with different densin molecules.

Response: We agree with the reviewer that we did somewhat overinterpret these data. We have now added experiments where we use activated (T286E mutant) α CaMKII to pulldown PP1 α in the presence or absence of coexpressed Densin-180. Here we see a strong increase in coprecipitated PP1 α when Densin-180 is present, which supports the idea that Densin-180 may scaffold a triple complex of these proteins (newly assembled supplemental Figure S9). In addition, we have made our wording regarding these experiments in the Results and Discussion sections more careful.

6. In the discussion, the authors conclude that “most of the binding to α CaMKII is lost upon deletion of the last 350 amino acids (from densin), but some specific binding remains, likely at the central site.” However, it does not appear that the activation of CaMKII was controlled in these assays, and it is important to note that the central site only binds to the activated kinase.

Response: we mentioned now that activation/phosphorylation of CaMKII is necessary for binding at the central site. In fact, we have used this (thanks to the reviewer for alerting us!) information as we performed coprecipitation of CaMKII, Densin and PP1 with the T286E mutant form of CaMKII, which mimicks the phosphorylation event (see above, Supplemental Figure S9).

7 The authors may wish to consider how the truncated densin proteins that may be expressed by some of the mutated DNAs relate to the numerous mRNA splice variants of the rat densin characterized previously (Jiao et al., 2008. PMID: PMC2814316).

Response: We thank the reviewer for alerting us to this issue, as it helped us to provide a possible explanation for our pedigree A patient. Two studies have addressed alternative splicing in rat and mouse *Lrrc7* genes (Jiao et al., 2008; Witte et al, PMC6690840). Frequent exon skipping events were observed for exons 17, 18, 22 and 23. Whereas none of our variants is localized in exons 17, 18 and 22, the Q1375* (pedigree A) and Asn1377Thrfs*13 (pedigree I) variants are located in exon 23. Based on RT-PCR data from the Witte et al paper, this exon is present in most transcripts at postnatal day 0, going down to about half of the transcripts at later stages of brain development. We have mentioned now in the results section which variants may be affected by this alternative splicing process. Importantly, pedigree A has two loss-of-function variants, and one of them may be spliced out.

8. I do not have the expertise to properly evaluate the patient and genomic data

Reviewer #2 (Remarks to the Author):

The manuscript by Willim and colleagues presents *LRRC7* as a new disease-gene. The authors provide sufficient clinical and molecular evidence to demonstrate that variants in *LRRC7* lead to the onset of a neurodevelopmental disorder mainly characterized by speech delay, intellectual disability, and aggressive behavior. The manuscript is well written and merits publication. Despite this interesting content, the manuscript would benefit from revisions. The following specific points need to be addressed:

- In the introductions, lines 92-93, the authors mentioned that a single patient with a loss-of-function variant in *LRRC7* was observed in an exome sequencing study of NDD patients (Popp 2017). However, there are two additional frameshift variants reported in HGMD that should be also mentioned: p.Lys120Valfs*2 (Yuen 2015, PMID: 25621899) and p.Val1032Serfs*5 (Fu 2022 and Zhou 2022, PMID: 35982160, 35982159)

Response: We thank the reviewer for alerting us to these cases. We have included the references and mentioned these cases at the appropriate position in the Introduction.

- In the introduction (line 94), the authors state that they collected a group of 30 patients. In the abstract (line 63), 31 patients are mentioned. The authors should address this discrepancy throughout the manuscript

Response: Again, thanks for notifying us of this mistake. We have redone the counting of cases, and in fact, there were 32, and now are 33 affected individuals with a confirmed variant in the *LRRC7* gene. We have corrected this throughout the manuscript.

- In the results paragraph "identification of *LRRC7* variants in patients with a neurodevelopmental disorders", the cohort identified from the 100 KGP and Genematcher are discussed separately. I understand that it makes sense to explain how the cohorts were assembled, but it would be less confusing if the patient mutations and clinical features would be discussed all together instead of separately. For example: "in total, we identified 31 patients with 24 different variants with the following clinical features..."

Response: we agree that this was confusing. Nevertheless, our intention was also to distinguish between a "discovery collection" of pedigrees where the initial statistical association was found; and

a “replication collection” where additional cases are found that support the statistical findings and expand the phenotypic spectrum. We have now entirely rewritten the “identification” part of the manuscript. Importantly, we have now created an expanded Figure 1 where all pedigrees are listed, together with the list of HPO phenotypes, again for all affected individuals.

- The authors speculate that the father of case 11 is likely carrier of the LRRC7 variant because both younger half siblings on the father’s side are mentally retarded (I would avoid using this term in the supplementary patients’ descriptions) and because the father displayed aggressive behavior and ID. However, in the absence of genetic testing in the father or in the two half siblings, it cannot be excluded that the reasons for the ID of the siblings and father might be different. If the authors want to provide an example of carrier parent with some degree of neurological manifestations, they should discuss a parent for whom genetic testing has been performed

Response: We have now eliminated this type of speculation. We had the chance to include one additional pedigree (Z) where indeed the parent carrying the pathogenic variant has quite a different phenotype (neuropsychiatric, but no ID) than the offspring, who shows the typical ID phenotype.

- At the moment, the clinical features of the patients are scattered across Figure 1, Supplementary Figure 1, the supplementary table and the supplementary text. I believe the manuscript would benefit if there would be a table summarizing the main clinical features of the LRRC7 patients and the total frequency of each feature (expressed in percentage) in the main text.

Response: we agree with this reviewer that it would be helpful and much more reader-friendly to provide an overall picture of the variants and phenotypes in one place. As mentioned above, we have now generated a larger Figure 1 which includes the initial statistical association (Fig. 1a), followed by both types of pedigrees listed from A to Z (Fig. 1b). In Fig. 1c we have now combined HPO terms for all pedigrees and also included a little bar graph which indicates the total frequency of each feature, as requested.

- Have the authors attempted to compare the clinical features of the patients with truncating variants with those of the patients with missense variants? Is there any difference that would help to phenotypically distinguish the two classes of patients?

Response. Currently, we have not identified a unifying feature of one group of patients. We have mentioned now that e.g. the hyperphagia phenotype is restricted to patients with a truncating mutation in exons 20 and 21. But there are others with stop mutations in this part of the mRNA who do not share this phenotype.

- The functional analysis of the missense variants in *C. elegans* (line 298) was only performed on mutations p.L211M and p.L296P. Is there any reason why specifically these two mutations were chosen and not the other four mutations in the LRR repeats (L65S, N94S, C106Y, and A139P)? Why was the analysis limited to these two mutations and not extended to the other mutations?

There were some practical reasons to limit the study to these two variants. The *C. elegans* studies were part of the NIH Undiagnosed Diseases Network (UDN), where two cases UDN334613, (p.Leu221Met), and UDN136306, (p.Leu296Pro), were nominated by the Duke UDN Clinical Site for model organism studies to provide functional data in support of a molecular diagnosis. Following modeling the two *LRRC7* variants, the *C. elegans* group moved on to modeling other candidate disease genes from undiagnosed patients in the UDN. The collaboration among the groups in this

manuscript, connected by GeneMatcher, began more than a year after the *C. elegans* studies were completed.

Based on conservation, two other residues would have been a good choice, namely N94 (identical in the *C. elegans* protein) and L65 (isoleucine in *C. elegans*). The others were not well conserved. We have mentioned the issue of sequence conservation in the text.

- I believe that the statement “Nonsense variants disrupt C-terminal interactions of Densin-180” (which is also the title of one of the results paragraph) is incorrect. As the authors themselves state, nonsense and frameshift variants are likely subject to nonsense mediated decay, which would lead to degradation of the mutant transcript. Hence, the truncated proteins are likely not produced in the cells. This applies also to the two tested mutations p.R1018* and p.Y1210* (both fall within exon 21 of 27 and are therefore likely associated with NMD). For this reason, the in vitro generation of the two resulting truncating proteins does not inform on the real pathogenic mechanism associated with truncated variants. This experiment exclusively shows that the PDZ domain of Densin-180 is important for the interaction with α -CaMKII and δ -catenin, which was previously already shown. Hence, the real pathogenic mechanism associated with truncating variants (frameshift and nonsense) is still unknown and can be investigated only in the presence of patients’ cells. Thus, the statement “Nonsense variants disrupt C-terminal interactions of Densin-180” is incorrect: the pathogenic mechanism of these variants is likely haploinsufficiency rather than due to impaired interaction

Response: this reviewer is of course correct. We have softened our statement here. We wish to note that these experiments are nevertheless useful for determining the relation between Densin-180 and the α CaMKII, where binding is partially lost here (see also comments by Reviewer 1).

- Why was the spine-shaft ratio of GFP-fluorescence intensities not measured for the mutations N94S, C106Y, and A193P (Fig 6)? I believe all the mutations should be assessed using the same parameters

Response: we have now included spine-shaft ratios for these three mutants, and in fact have tried to make sure that all relevant parameters are measured for all variants, with the exception of the *C. elegans* data. These new data are incorporated at the appropriate positions and have also been summarized in a Table at the end of the text.

- In supplementary Figure S6, two different clones of MACF1 are used. What is the difference between the two clones? Is the cloned fragment exactly of the same size in both clones? Were the plasmid sequences verified by Sanger? The authors should also specify the size of the MACF1 fragments used (from amino acid XX to amino acid XX of MACF1).

We thank this reviewer for pointing this out; we resequenced these clones, and noted that clone 1 was contaminated with another GFP-containing plasmid. Therefore we eliminated these blot lanes from the Figure. The residues obtained in the yeast two hybrid screen for both MACF1 clones and for the single β -spectrin clone are now listed in the Supplemental data.

- Several analyses have been performed to assess the pathogenicity of the missense variants within the LRR domain (functional analysis in *C. elegans*, dendrite morphology, post synaptic targeting, interaction with α -CaMKII, δ -catenin, Shank3, MACF1, Erbin, PP1 α). I think the manuscript would benefit if there would be a table in the main text summarizing the analyses done and specifying which analysis/parameter is altered for each specific LRR-missense variant. In this way, it would

become immediately apparent which specific Densin-180 function is altered in the presence of each mutation

Response: this is a very helpful suggestion, thank you. As mentioned before, we have now included such a Table.

Minor comments:

- In Figure 6, the legend letter D is not visible in the figure

Response: this has been fixed.

- In the legend of Figure 4, specify threshold for significance (***, significantly different from wt: which is the threshold?)

Response: this has been fixed.

- In Supplementary Figure 5, the mutation L221M is wrongly indicated as L221P in graphs B, D, and F.

Response: this has been fixed in what is now Supplemental Figure 7.

In addition, in the Western blot of part A, the band corresponding to L221M is not detected, neither in the input, nor in the immunoprecipitated fraction. Do the authors have an additional western blot picture where also L221M signal is visible? How many times was the analysis performed? (n=3?)

Response: Indeed, as observed by this reviewer, expression levels were variable for different variants in this type of experiment. We have now performed additional experiments to add to this Figure, and in particular have now included a fourth repeat, providing a better blot and improving the statistics of this experiment.

Reviewer #3 (Remarks to the Author):

This is a well written study on association of the LRRC7 deficiency with human disease where intellectual disability, neurodevelopmental delay and abnormal behavior are some of the common features. In total, 31 individuals were identified with heterozygous (predominantly loss of function) variants in LRRC7. Further molecular studies showed that some of the (patient) missense variants in the PDZ domain interfere with targeting densin to neurons, while variants in the LRR domain interfere with binding of the LRRC7 interaction partners that were identified in this manuscript. Modeling using *C. elegans* model organism showed that one of the missense variants (L248P), identified in patients, is pathogenic (i.e. leads to embryonic lethality). Overall, this is an extensive cohort of patients where various molecular assays were conducted to confirm the role of LRRC7 in neurodevelopment. The work is important and of good quality.

Comments to the author:

Major comments

- In terms of the patient cohort, it is difficult to follow which patients make the cohort of 31 described here; how many come from which study and how many belong to the same family. There are 21 case reports in the supplemental and 21 in the supplemental Excel table; it would be helpful to explain it a bit clearer how this number relates to 31.

Response: we are sorry that we failed to count our patients correctly; there are now 33 patients in total. We agree that the different cohorts were difficult to follow in the first version of the manuscript (see above, comments by Reviewer 2). We have now completely rewritten the initial part of the Results section, and combined patients from our two cohorts in Figure 1. As clearly described

now, there are 11 patients from the 100k genomes study, which form our initial “discovery collection” (pedigrees A-I). The second group of now 23 cases (pedigrees J-Z, then #) forms the “replication collection” where additional cases are found that support the statistical findings and expand the phenotypic spectrum (see also above, comments for reviewer 2). Clinical data for these 23 cases are presented both in the supplement and in the supplemental Excel table.

- Detailed description and main text discussion is necessary for the families where variants are inherited from “unaffected” parents. Do these variants tend to be less severe, missense? Do phenotypes in these families tend to be milder? Discussion on potential reduced penetrance observed with other disorders related to LRRC7 would strengthen the manuscript.

Response: We do not observe that inherited variants cause a less severe phenotype. We have added a paragraph in the Discussion related to this.

- Lines 272-274 “This is exemplified ...” It is not clear what is the basis of the conclusion that father who is unavailable for testing is likely a carrier. Please explain and provide evidence.

Response: We have now removed this discussion for case 11; in addition, as we had the new case of pedigree Z, we mentioned the phenotypic variability evident here by mother and daughter sharing the same variant, with a very different phenotypic outcome.

- Considering high phenotypic variability between patients, it is necessary to include information on other variants in the exomes/genomes. Were exome/genome data checked for other contributing variants (i.e. dual hits/diagnoses), especially in more severe cases?

Exomes are routinely checked for other variants; variants which appeared to be relevant are listed in the Supplementary Table 1 with the respective patients.

- In terms of *C. elegans* work, it is unclear why the authors used the *udn71* allele rather than generating a targeted knockout (complete ORF deletion). It seems that the *udn71* allele was generated as a byproduct of CRISPR, unintentionally. It is bit worrisome how the *udn71* was obtained as this genome may contain additional unintentional variants. It would be preferential to obtain a deletion allele where entire ORF is deleted.

Response: Legouis et al. (2000) demonstrated that homozygotes for the multi-locus deletion *sDf35*, premature stop alleles *s1431* and *s1451*, splice site allele *s1455*, and strong RNAi knockdown, all result in 1.5-fold embryonic arrest, which the authors concluded was the null or very strong loss of function phenotype. The *udn71* deletion allele and all the variant and control edited strains were generated in the VC2010 genetic background, and thus more comparable than the legacy alleles isolated in the N2 genetic background. Since the *udn71* deletion allele has the same phenotype as the known null or very strong loss of function alleles, it was considered to be sufficient for the current studies.

- Also, why focus only on the p.L211M and p.L296P variants for modelling in *C. elegans*? Were these the only variants conserved well enough to allow for the knock-in CRISPR?

Response: See response to a similar question from Reviewer #2. One reason was that not all of the mutated residues in patients are highly conserved in *C. elegans*. The other reason was that *C. elegans* studies had already been concluded, when additional mutations were found.

- Were any attempts made to further characterize the L173M (L211M) variant? Is it temperature sensitive? Any subtle defects using AJM::GFP? How about seam cell phenotypes?

let-413 L173M homozygotes showed no obvious effect on brood size, growth rate, body size, thrashing or crawling movement phenotypes, larval/adult spermathecal morphology defect or temperature sensitivity. Additionally, *let-413* L173M/ *let-413(udn71)* compound heterozygotes showed no thrashing or crawling movement or viability phenotypes. We cannot rule out that there were subtle AJM::GFP or seam cell phenotypes that were missed.

- In addition to *C. elegans* work, the L211M when tested using other methods does not seem to show any obvious effects. How did the authors then conclude that L211M is mild rather than benign variant? The same question also applies to A193P (line 400).

If *C. elegans* experiments with a gene-variant gives a positive result, i.e., there are observed phenotypes, then the variant is damaging in the worm, and likely damaging in humans. However, a negative result, i.e., not observing a phenotype, is in our view interpreted as non-informative, rather than leading to the conclusion that the variant is benign. This is because we may not have tested the appropriate phenotype, may not have examined under the appropriate conditions, or may not have used an assay that was sufficiently sensitive. If there were only data from *C. elegans*, then *let-413* L173M/ *LRRC7* p.L221M would be considered a VUS. However, for the A193P variant there was a significant loss of binding for at least one of the interaction partners tested (Figure 8). Therefore we have classified it as mild. As, after all these tests, there was not a single significant effect for the L221M variant, we have now reclassified it as a VUS.

- Given that different molecular tests were done on different *LRRC7* variants, it is necessary to include a summary table to summarize for how many of the variants were validation tests attempted, for how many not, for how many only one test, for how many the experimental tests indicate potential effect and for how many benign status. It would be also useful to use ACMG guidelines to classify the variants. Will the variants be submitted to ClinVar?

We have now included a summary Table for all variants/all tests. Variants will be submitted to the appropriate database upon publication of the manuscript (LOVD).

- Also, I could not find anywhere in the manuscript population frequency of these variants (e.g. in gnomAD and TopMed databases), please include.

Variants identified here were not detected in the gnomAD database. We have included a statement about this in the Results section.

- In terms of phenotype, it seems that the single overlapping feature between the patients is abnormal nervous system physiology/abnormality of the nervous system, but it is not explained what that is and what kind of testing was used to determine this phenotype.

Response: there are several overlapping phenotypes, among them “intellectual disability”, which has been mentioned throughout the text. Abnormality of the nervous system is the wider term, which obviously includes intellectual disability.

- Hyperphagia does not appear to be included in the graphic presentation of phenotypes.

Response: this is included in “abnormal eating behaviour, which may be the more general term.

Minor comments

- Genematcher is misspelled throughout, please revise to GeneMatcher.

This has been fixed

Reviewer #4 (Remarks to the Author):

This study describes the discovery and characterization of rare variants in the *LRRC7* gene, which codes for the LAP protein Densin-180, and their association with a neurodevelopmental disorder in humans. The study combines data from the 100k genomes project with patient data from genetic laboratories. In total, 31 individuals with *LRRC7* variants are identified, showing a core phenotype of intellectual disability, delayed language development, ADHD, autism, and seizures in some cases. Aggressive or impulsive behavior is also common among these patients.

The functional impact of loss-of-function and missense variants affecting the *LRRC7* gene are explored using biochemical, genetic (*C. elegans*), cellular, and imaging approaches. Patient variants in *LRRC7* variants disrupt interactions with known partners, such as δ -catenin, and newly identified interacting proteins (Erbin and MACF1). PP1 α is also identified as an important interaction partner of Densin-180, indicating a potential role in regulating synaptic signaling. The authors discuss the importance of the Densin-180/PP1 complex in synaptic plasticity and the potential impact of *LRRC7* variants on this process.

Major comments:

- Regarding the functional validation studies in *C. elegans*:
 - Patient variants are heterozygous. The authors indicate that *let-413* mutants are recessive. Are they fully recessive or are there also hints of haploinsufficiency in *C. elegans*?

Response: Haploinsufficiency in *C. elegans* is thought to be rare (Hodgkin, J. (2005) Karyotype, ploidy, and gene dosage. WormBook, 1–9. <https://doi.org/10.1895/wormbook.1.3.1.>), likely because gene expression is titrated differently in different species. Given that *LRRC7* has a pLI=1, the genetic model tested in *C. elegans* is if the missense variant is partial or complete loss of function, which is assessed in the homozygous state. We did not observe an obvious phenotype in *let-413(udn71)* heterozygotes but cannot rule out that we missed a subtle phenotype. Dominant phenotypes have not been reported for *let-413* mutations in the literature.

- Along these lines, the authors conclude that L221M is a mild mutation based on very limited impact in different assays and no obvious impact in *C. elegans*. Maybe some subtle perturbations of the AJM-1::GFP pattern could provide confirming evidence about the pathogenicity of this variant.

- In fact, in the absence of stronger evidence about its functional impact, wouldn't it be prudent to reclassify the variant as VUS, pending additional functional data? This conclusion may have important clinical implications for the patient, potentially prompting further genetic analysis.

Response: See response to similar question from Reviewer #3. We have now reclassified this as a VUS.

- Statistical analysis of Western/IP quantifications.

Student's t-tests are applied in Figures 4B,8B,8D,8F,S2,etc., to compare the quantified Western/IP data. Applying a Student's t-test to this data is not possible, as there are only three data points. Furthermore, post-hoc multiple-comparisons tests also need to be applied to correct these analyses. Please correct these issues throughout the manuscript.

We apologize for being somewhat short and unprecise on the statistics. We have now added additional experiments/datapoints where necessary. In addition, we have clarified better which statistical tests were used, and which multiple comparison tests were included (mostly: Dunnett's) after the initial ANOVA testing.

- Regarding Figure 4A,B:

- It is unclear whether the three samples presented in Figure 4A are independent biological replicates. Please provide this type of information throughout the manuscript, whenever applicable.

Response: Three independent biological replicates were analysed on a single Blot. In the meantime we have performed additional (independent) experiments; so the n=4 independent experiments for this experiment now.

- Regarding Figure 4C-F:

- The distribution of Shank3 appears very different in the L1567Q mutant. Is there data ensuring that synapse formation is not altered in general upon L1567Q expression in this cellular context? Indeed, an alternative hypothesis could be that the over-expression of the L1567Q mutant alters the synaptic organisation in general or the distribution of Shank3 specifically.

Response: we have now included data on the number of Shank3 clusters (i.e. number of synapse) for all variants in the new supplemental Figure S6, showing that none of the mutant has any effect on this parameter. In addition, we have changed the Neuron which is shown in Fig. 4c, to make it more representative.

- Regarding Figure 4C-F and Figure 6:

- Methodological information is very limited regarding these experiments. For instance, how are Densin-180 clusters counted? From how many independent experiments is the data derived (independent transfections, cells, dendrites, etc.). Reading the Methods section, it seems they are counted "manually". What kind of image treatment was applied (e.g., thresholding, segmentation) to determine whether a group of pixels indeed constitutes a "synaptic cluster"?

Response: Additional information on microscopic work and evaluation has now been included in the Methods section, and in the Legends to the respective Figures..

Regarding the enrichment of Densin-180 at Shank-3-labeled sites, Pearson correlation analyses are generally used to quantify synaptic co-localisation patterns. See for example Verstraelen P, et al., iScience. 2020). Please implement this type of analysis to determine enrichment (or not) of Densin-180 at Shank3 labeled clusters for wild-type and patient variants.

We have followed the advice of this reviewer and performed Pearson correlation analysis using the Jacop plugin of the ImageJ software. Indeed, we observed a reduced colocalization with Shank3 for some of the variants, as shown no in Supplemental Figure S5.

Minor comment

- Fig.2B : The letters in the "consensus" line are not aligned with the sequence of the different LRR repeats.

Response: this has been fixed

Reviewer #5 (Remarks to the Author):

The authors describe 31 individuals with variants in the LRRC7 gene and significant neurodevelopmental anomalies. This is a major paper defining a new human disease going well beyond describing the phenotype and genotype, to also defining mutation mechanisms and enlarging our knowledge of LRRC7 functions.

The paper is well written, concise, and clear. I do not think it should be shortened.

I have no problems with the descriptions, experiments, discussion, or data; below are some minor suggestions to consider.

Introduction

Good and interesting.

Abstract says 31 patients, Intro says 30? This is a confusing/annoying. Maybe better later stated as, "In total we describe 31 individuals with variants in LRRC7.", as one is from the literature.

Response: As mentioned in response to the other reviewers, we had some counting issues. Now we are at 33 affected individuals, and have corrected this throughout the text.

Materials and methods

I cannot comment on the C. Elegans work methods or results.

The HEK 293T cell line is called both 293T cells and 293T HEK – please harmonise, e.g. line 547 in Figure legends.

Response: This has now been harmonized to "293T cells"

Results

"In many cases this is associated with delayed speech and language development as well 253 as delayed fine motor development. Several patients were affected by autism spectrum disorder (ASD) 254 as well as ADHD. Furthermore, selective mutism and impulsive and aggressive or auto-aggressive 255 behaviour was reported..." – could numbers with each feature be shown in brackets? Later such numbers are given in the paragraph, e.g. for brain malformations.

Response: The frequency of these phenotypes are now shown in Figure 1c. There is a little bar graph between HPO terms and the phenotype boxes for the patients.

"Four patients with truncating 256 variants in the central part of the mRNA/protein showed hyperphagia due to reduced satiety, leading 257 to class II obesity in three of them." Were these the only patients with such variants - if so, can that be stated please.

Response: There were more patients with truncating variants, many of whom did not show hyperphagia. The relation between hyperphagia and genotype is therefore not of high penetrance. Nevertheless, we have noted that hyperphagia is limited to patients with deletions in exon 20 or 21.

"Again, loss-of-function variants (nonsense, splice site and frameshift variants, and a 99 kB deletion 261 encompassing the last exon), were observed (13 patients)." Again is not needed.

Response: This part has been altered, due to the restructuring of the initial parts of the Results section. So the “again” is gone.

It is cumbersome that the phenotypes and genotypes of each group is described separately, and not collectively. Whilst I understand why the paper was written this way, combining the results would be much easier for the clinician or scientist who will read the paper.

Response: obviously this is a concern shared by several reviewers. We have responded by combining all pedigrees and all patient phenotypes in Fig. 1. This should make it much easier to follow.

“Seven pedigrees were identified with inherited cases, indicating that LRRC7 variants display incomplete penetrance.”

May be better to be,

“Seven pedigrees were identified with inherited cases, indicating that LRRC7 variants may display incomplete penetrance.”

Response: We have changed this.

Paragraph from lines 280 to 297; the descriptions of the amino acid changes are valid (and interesting to this Reviewer), but inclusion of a known score of pathogenicity, say from AphaSense, may enhance readability.

Response: there is, for each variant, the CADD score present in the clinical Table.

Line 400 “C. elegans embryo (Fig. 3). Thus, these two appear to be rather mild mutations.” Did this coincide with a milder phenotype?

Response: as we did not observe a phenotypic change for the L221M variant in any of our functional tests, we have now reclassified it as a VUS, though this patient does not have a particularly mild phenotype.

Discussion

Line 418, ...of the 100k genomes project... you have previously defined this as 100KGP, so why not use the abbreviation?

Response: this has been changed.

Line 92. “In addition, a single patient with a loss-of-function variant in LRRC7 was observed in an exome sequencing study of NDD patients (20).”

To

“In addition, a single patient with a loss-of-function variant in LRRC7 was observed in an earlier exome sequencing study of NDD patients (20).”

Response: this has been changed

Line 426 “Furthermore, aggressive or impulsive behaviour is prevalent in many cases reported here.” Could you comment on any sex effect of this behaviour in the 30 cases, and state in how many cases this could be judged, i.e. if very young, aggression maybe hard to assess.

Response: we have added numbers here for the replication cohort where more phenotypic detail is available. Out of 22 cases (one of them is too young), 9 are listed with impulsivity or some form of aggression (6 males, 3 females).

Line 440, "We noted that in several cases, these variants were inherited from healthy parents, indicating incomplete penetrance of the phenotype. However, due to the complexity of the phenotype, it appears possible that specific aspects of e.g. aggression or obesity may have been overlooked in the parents."

It was unclear how many parents had been properly/completely phenotyped, and this should be stated before any comment is valid about incomplete penetrance. A more nuanced approach maybe to state that it is unclear if the condition always manifests, and omit the last sentence (as it could also apply to learning difficulties in childhood, without interviewing the respective grandparents.

Response: we followed the advice of this reviewer and used the more nuanced approach.

Line 444 " Further insight into the functional aspects of Densin-180 required for normal brain function may be.." may be changed to "were"

Response: this has been changed

Line 450 "...which were not known so far." Delete "so far"

Response: this has been changed

Line 533 Figure, "AlphaFold" not "alphaFold".

Response: this has been changed

Line556 "(F). ***, significantly different from WT, t-Test." *** is not defined, and is $p < 0.001$? (Embarrassingly I looked this up, and it is either "Student's t test" or "T test", not t-Test.)

Response: this has been changed

Supplementary data

The case histories are a valuable addition.

REVIEWER COMMENTS

Reviewer #1 (Remarks to the Author):

I continue to think that the manuscript describes some interesting patient LRRC7 mutations (encoding densin), that are associated in some way with neurodevelopmental disorders. However, the authors conclusion "that LRRC7 encodes a major 74 determinant of intellectual development and behaviour" is not really justified based on the data presented. While some of the data are suggestive of functional alterations in densin, they are very preliminary in nature and their biological relevance is unclear, as detailed below. Taken together, the limitations compromise the significance and impact of the manuscript.

Comments on author responses to my original concerns.

Point 1:

To be perhaps more explicit, I was not asking the authors to conduct the screening studies in neurons, but only to provide complementary data supporting the interpretation that some of the novel interaction partners that were detected are indeed associated with densin in the brain. I completely understand the technical solubility issues associated with conducting co-IP, BioID and related protein-protein interaction studies in neurons from our own work. Although these issues certainly introduce caveats, their implementation has provided innumerable insights over the years. Frankly, the synaptic colocalization studies may be consistent with an interaction, but they are very weak evidence for a direct interaction. In my view, the lack of additional data supporting direct interactions with densin substantially reduces the significance and impact of the work identifying novel interaction partners.

Point 2:

Thanks for fully addressing this issue in the revised manuscript.

Point 3:

Thanks for including a more extensive data set investigating potential effects of the patient mutations on neuronal morphology. These studies appear to have been performed carefully. The Sholl analysis failed to provide truly convincing effects on dendritic complexity, and it is appropriate to include these largely negative data in the Supplement (Fig. S4). Similarly, the more rigorous analyses of postsynaptic clusters also failed to reveal significant effects of the patient mutations (Fig. S6). It is unfortunate that these efforts yielded largely negative results, making it hard to understand how the differences in clustering of the densin variants (Figs 5 and 6) might impact neuronal function and thereby behavior.

Point 4:

Thanks for fully addressing this issue in the revised manuscript.

Point 5:

The new experiments examining the possibility of a tripartite CaMKII-densin-PP1 complex (Fig. S9) are not particularly compelling. Regarding panel A of this figure, the authors continue to assert: "Note that co-expression of kinase or phosphatase does not affect the level of phosphatase or kinase bound to Densin-180, suggesting that both proteins can bind at the same time to Densin-180", but this is only true if they can demonstrate that they have isolated stoichiometric complexes. As currently presented, the lack of effect may simply result from sub-stoichiometric binding of CaMKII or PP1 to different subpopulations of densin. In panel B, I do not understand how the electrophoretic mobility of the RFP-CaMKII fusion protein is about 55 kDa, which is the same size as the untagged protein – was the RFP cleaved from the expressed protein? Furthermore, I do not find the stated interpretation that densin increases the association of PP1 with the mutated CaMKII particularly convincing, notwithstanding the quantification shown, due the large variation on the values in the presence of

GFP-densin. The authors may find it better to normalize the levels detected with GFP present to the levels detected with GFP-densin present, and then average the normalized ratio across all four (independent biological?) replicates before conducting a one-sample t test. The authors should also compare the association of WT and mutated CaMKII with PP1, in the absence and presence of densin. These issues make it hard to understand the significance of the impact of several densin mutations on PP1 binding, not only for reasons discussed in point 1, but also because these studies were conducted with the isolated densin-LRR domain, not the full length densin protein. Moreover, it is notable that the mutations similarly affect the binding of both Erbin and PP1 to the densin-LRR domain, but there is no explanation of how/why. In addition, the lower expression levels of these mutants indicates that they may be much less stable proteins. Do these mutations affect the folding of the densin-LRR domain?

Point 6:

Thanks, although it is worth noting that the T286E mutation does not fully mimic the effect of autophosphorylation.

Point 7.

Thanks for addressing this issue in the revised manuscript.

Reviewer #2 (Remarks to the Author):

The authors have addressed all my concerns related to the previous version of the manuscript.

There are only few additional minor comments from my side:

- Table S1 is mentioned only in the result paragraph "LRR domain variants interfere with binding to novel interaction partners of Densin-180" and in the discussion, but it is not mentioned in the paragraph "Identification of LRRC7 variants in patients with a neurodevelopmental disorder", where the clinical features of the patients are presented. On the other side, the case reports descriptions found in the supplementary material are never mentioned across the entire manuscript. Thus, the first paragraph of the results would benefit of a sentence that specifies that "detailed clinical data can be found in the supplementary Table S1 and in the case descriptions"

- Why are cases of pedigrees A-I not included in the Table S1, nor in the supplemental case reports descriptions?

- Throughout the manuscript, the term loss-of-function variant is used as a synonym for truncating variant. However, missense variants are also shown to lead to loss-of-function due to impaired interactions. For this reason, it is more appropriate to refer to splicing, nonsense, and frameshift variants as "truncating variants" rather than loss-of-function variants

- When referring to gnomAD, please indicate which version (i.e. v4.0.0 or v2.1.1)

- The bar graph as a representation of the percentage of each clinical feature in Figure 1c is a nice addition. This reviewer believes however that the percentage of the main features should also be clearly indicated in the text

Reviewer #3 (Remarks to the Author):

This is a resubmitted and revised version of the manuscript that describes association of the LRRC7

deficiency with human disease (marked by intellectual disability). The number of patients included in the study is recounted to now include 33. Overall, the authors made an excellent attempt at responding to reviewers comments and suggestions. I only have a couple of additional comments.

1. Please indicate which version of gnomAD database was used to search for variants. Was TopMed BRAVO database used as well?
2. Please consider phrasing all "patient mutations" as "patient variants", the manuscript uses these two terms interchangeably, which is confusing.
3. Please review the manuscript for errors:
 1. line 68 - "in" should not be italicized
 2. line 373 - "This in agreement ..." change to "This is in agreement ..."
 3. line 386 - " Wild type and variants ..." please state which variants

Reviewer #4 (Remarks to the Author):

I thank the authors for their precise answers to my questions and for the careful revision of their manuscript. I have no further comments or requests at this time.

Reviewer #5 (Remarks to the Author):

My comments and queries have all been addressed.

Response to reviewers.

We would like to thank all five reviewers for their constructive comments to our manuscript, which has helped to improve it a lot. We hope that in this submission, we could clarify all critical points to make it ready for acceptance.

Reviewer 1

Point 1:

To be perhaps more explicit, I was not asking the authors to conduct the screening studies in neurons, but only to provide complementary data supporting the interpretation that some of the novel interaction partners that were detected are indeed associated with densin in the brain. I completely understand the technical solubility issues associated with conducting co-IP, BioID and related protein-protein interaction studies in neurons from our own work. Although these issues certainly introduce caveats, their implementation has provided innumerable insights over the years. Frankly, the synaptic colocalization studies may be consistent with an interaction, but they are very weak evidence for a direct interaction. In my view, the lack of additional data supporting direct interactions with densin substantially reduces the significance and impact of the work identifying novel interaction partners.

Response: We have now attempted to be more careful in our description and especially discussion of our findings, and have in particular moderated our conclusions with respect to the PP1 α / α CaMKII signalling complex mediated by Densin-180. We are aware that further functional assays in vivo (i.e. in ko mice, or better in Crispr-generated mice carrying relevant missense variants), will be necessary to substantiate our findings. However, this would be beyond the scope of the current manuscript.

Point 2:

Thanks for fully addressing this issue in the revised manuscript.

Point 3:

Thanks for including a more extensive data set investigating potential effects of the patient mutations on neuronal morphology. These studies appear to have been performed carefully. The Sholl analysis failed to provide truly convincing effects on dendritic complexity, and it is appropriate to include these largely negative data in the Supplement (Fig. S4). Similarly, the more rigorous analyses of postsynaptic clusters also failed to reveal significant effects of the patient mutations (Fig. S6). It is unfortunate that these efforts yielded largely negative results, making it hard to understand how the differences in clustering of the densin variants (Figs 5 and 6) might impact neuronal function and thereby behavior.

Response: Obviously, we, and in particular the co-worker doing the experiments, would have liked to see more substantial effects in these experiments. However, the fact that brain development and formation of synapses is almost completely normal in the complete absence of LRRC7, should tell us that the major role of Densin-180 is not in neuronal morphology. In contrast, synaptic signalling and plasticity is altered in the ko mice, suggesting that this is where we have to look for the neuronal effects of Densin-180 variants.

Point 4:

Thanks for fully addressing this issue in the revised manuscript.

Point 5:

The new experiments examining the possibility of a tripartite CaMKII-densin-PP1 complex (Fig. S9) are not particularly compelling. Regarding panel A of this figure, the authors continue to assert: "Note that co-expression of kinase or phosphatase does not affect the level of phosphatase or kinase bound to Densin-180, suggesting that both proteins can bind at the same time to Densin-180", but

this is only true if they can demonstrate that they have isolated stoichiometric complexes. As currently presented, the lack of effect may simply result from sub-stoichiometric binding of CaMKII or PP1 to different subpopulations of densin.

Response: We agree that this experiment is not very convincing and have removed it. The rest of this Figure is now Figure S10, including some new experiments.

In panel B, I do not understand how the electrophoretic mobility of the RFP-CaMKII fusion protein is about 55 kDa, which is the same size as the untagged protein – was the RFP cleaved from the expressed protein?

Response: we thank this reviewer for pointing out this mistake. Assignment of marker bands was wrong here. We have corrected this, and confirm that the mRFP-CaMKII fusion indeed runs slightly above the 72 kDa marker, which roughly corresponds to the molecular weight.

Furthermore, I do not find the stated interpretation that densin increases the association of PP1 with the mutated CaMKII particularly convincing, notwithstanding the quantification shown, due the large variation on the values in the presence of GFP-densin. The authors may find it better to normalize the levels detected with GFP present to the levels detected with GFP-densin present, and then average the normalized ratio across all four (independent biological?) replicates before conducting a one-sample t test.

Response: we thank the reviewer for the different proposals to improve the presentation of our data. However, we felt that we should not do the normalization of the GFP-Densin data to the GFP only-values (or the other way round), as we would lose variability (control would be 1 in each experiment) thereby precluding any statistical analysis. Instead we have now included a graphical representation of the individual experimental pairs of four independent biological repeats. This makes it more clear that in each experiment, the increase in formation of a PP1/ α CaMKII complex brought about by the presence of Densin-180 is more than 2 fold. These data are now in Figure S10.

The authors should also compare the association of WT and mutated CaMKII with PP1, in the absence and presence of densin.

Response: we have included the comparison between WT and mutant CaMKII. The amount of GFP-densin-180 which precipitates with the WT CaMKII is strongly reduced when compared with the mutant. Nevertheless, for both variants of the kinase, Densin-180 induces a significant increase in the amount of PP1 which associates and coprecipitates with the mRFP-tagged kinase. This may indicate that formation of a possible triple complex is largely driven by the C-terminal binding site, which does not require activation of the kinase. However, in the absence of further stoichiometric information we did not elaborate on this further in the main text. These data are now included in supplemental Figure S10 C,D.

These issues make it hard to understand the significance of the impact of several densin mutations on PP1 binding, not only for reasons discussed in point 1, but also because these studies were conducted with the isolated densin-LRR domain, not the full length densin protein.

Response: we wish to note here that the experiments addressing a possible triple complex were performed with full-length Densin-180 and not the isolated LRR domains. This was of course necessary, as the CaMKII binds outside the LRR domain at two sites which are more C-terminal. We have now added a new Figure S9, where we show that binding of PP1 to full length Densin-180 is affected in a similar way by the mutations, as was observed before for the LRR domain only (L65S and N94S were tested).

Moreover, it is notable that the mutations similarly affect the binding of both Erbin and PP1 to the densin-LRR domain, but there is no explanation of how/why. In addition, the lower expression levels of these mutants indicates that they may be much less stable proteins. Do these mutations affect the folding of the densin-LRR domain?

Response: We have now added some Discussion on this topic. Densin-180/LRRC7 is not the only protein where the LRR repeat region has been recognized as an interaction partner for the catalytic subunit of PP1. More detailed knowledge is available for SDS22/PPP1R7, which appears to serve as a repository for metal-deficient, inactive PP1; and for Shoc2, which serves as a scaffold for MRAS and PP1 in the Ras/Raf/Map kinase signalling pathway. In SDS22 and in the published Shoc2 structures, PP1 covers a large surface on the concave side of the LRR structure involving several of the repeats. In Shoc2, MRAS also binds on the concave side in tight association with PP1. Thus, the curvature of the LRR domain seems important to establish this interacting surface. We think that this is altered for Densin-180 by the different mutations, as these are likely to interfere with the tight packing of the repeats. Also, the hetero- or homodimerization (with Erbin, or with Densin-180) likely occurs via the concave surface, as it was shown to do so in Scribble. Importantly, the Shoc2-based structures appear as the closest “hits” when homology modelling of the Densin-180 LRR region is performed e.g. on the Swissmodel server. This could mean that additional proteins bind side-to-side with PP1 to the Densin-180 LRRs. Identifying such additional interactors will be an important avenue for further research.

Point 6:

Thanks, although it is worth noting that the T286E mutation does not fully mimic the effect of autophosphorylation.

Point 7.

Thanks for addressing this issue in the revised manuscript.

Reviewer #2 (Remarks to the Author):

The authors have addressed all my concerns related to the previous version of the manuscript.

There are only few additional minor comments from my side:

- Table S1 is mentioned only in the result paragraph “LRR domain variants interfere with binding to novel interaction partners of Densin-180” and in the discussion, but it is not mentioned in the paragraph “Identification of LRRC7 variants in patients with a neurodevelopmental disorder”, where the clinical features of the patients are presented. On the other side, the case reports descriptions found in the supplementary material are never mentioned across the entire manuscript. Thus, the first paragraph of the results would benefit of a sentence that specifies that “detailed clinical data can be found in the supplementary Table S1 and in the case descriptions”

Response: We thank the reviewer for this comment and suggestion, and have in fact added this sentence at the appropriate position in the Results section.

- Why are cases of pedigrees A-I not included in the Table S1, nor in the supplemental case reports descriptions?

Response: The phenotype data for the 100KGP participants included in our association analysis (which includes members of pedigrees A-I) were collected as Human Phenotype Ontology (HPO) terms by National Health Service clinicians independently of the other non-100KGP study participants in this paper (pedigrees J-#). Consequently, we do not have access to phenotype information for the 100KGP cases in pedigrees A-I in the form presented in Table S1 and in the case report descriptions. We only have access to HPO data for these cases. In our revised manuscript, we have provided an additional table showing all the HPO terms attached to the LRRC7 variant carriers in pedigrees A-I available through the Genomics England Research Environment (new Supplemental Table S2).

- Throughout the manuscript, the term loss-of-function variant is used as a synonym for truncating variant. However, missense variants are also shown to lead to loss-of-function due to impaired interactions. For this reason, it is more appropriate to refer to splicing, nonsense, and frameshift variants as “truncating variants” rather than loss-of-function variants

Response: we have changed this for most occurrences.

- When referring to gnomAD, please indicate which version (i.e. v4.0.0 or v2.1.1)

Response: see below, response to Reviewer 3. We have now reanalysed our variants with gnomAD v4.1.0 and have indicated this in the manuscript.

- The bar graph as a representation of the percentage of each clinical feature in Figure 1c is a nice addition. This reviewer believes however that the percentage of the main features should also be clearly indicated in the text.

Response: Main features have now been listed with percentages in the results text

Reviewer #3 (Remarks to the Author):

This is a resubmitted and revised version of the manuscript that describes association of the LRRC7 deficiency with human disease (marked by intellectual disability). The number of patients included in the study is recounted to now include 33. Overall, the authors made an excellent attempt at responding to reviewers comments and suggestions. I only have a couple of additional comments.

1. Please indicate which version of gnomAD database was used to search for variants. Was TopMed BRAVO database used as well?

Response: we did not use TopMed Bravo. We have reanalysed our variants with gnomAD v4.1.0 (now indicated in the manuscript) and have detected one of the truncating frameshift variants (pedigree U) in this database with two cases. All other variants were not detected.

2. Please consider phrasing all “patient mutations” as “patient variants”, the manuscript uses these two terms interchangeably, which is confusing.

Response: This has been changed throughout the text

3. Please review the manuscript for errors:

1. line 68 – “in” should not be italicized

2. line 373 – “This in agreement ...” change to “This is in agreement ...”

3. line 386 – “ Wild type and variants ...” please state which variants

Response: Thanks for paying attention to these details. These errors, and several more, have been corrected.

Reviewer #4 (Remarks to the Author):

I thank the authors for their precise answers to my questions and for the careful revision of their manuscript. I have no further comments or requests at this time.

Reviewer #5 (Remarks to the Author):

My comments and queries have all been addressed.